# Pre-stimulus alpha power modulates trial-by-trial variability in theta rhythmic multisensory entrainment strength and theta-induced memory effect
Danying Wang [1,2] ✉, Eleonora Marcantoni [2], Kimron L. Shapiro[3] & Simon Hanslmayr [2] ✉

Binding multisensory information into episodic memory depends partly on the timing of the hippocampal theta rhythm which provides time windows for synaptic modification. In humans, theta rhythmic sensory stimulation (RSS) enhances episodic memory when the stimuli are synchronised across the visual and auditory domain compared to when they are out-of-synchrony. However, recent studies show mixed evidence if the improvement in episodic memory is the result of modulating hippocampal theta activity. In the current study, we investigated whether pre-stimulus brain state could explain part of this variance in the neural and behavioural effects induced by the RSS, via recording 24 participants' brain activity with MEG during a multisensory theta RSS memory paradigm. Our findings suggest that pre-stimulus alpha power modulates entrainment strength in sensory regions, which in turn predicts subsequent memory formation. These findings suggest that for non-invasive brain stimulation tools to be effective it is crucial to consider brain-state dependent effects.

Episodic memory provides humans with the ability to mentally travel back to the past[1] where experiences typically involve associations between multimodal information, such as a concert you have seen. Binding the arbitrary sensory stimuli into a coherent memory episode depends on the hippocampus, which receives multimodal inputs from corresponding neocortical areas[2,3]. Moreover, hippocampal synapses are more likely to undergo activity-induced synaptic modifications such as long-term potentiation (LTP) and long-term depression (LTD)[4,5]. Evidence from rodent literature suggests that whether LTP or LTD occurs depends on whether stimulation is delivered at the peak or the trough of the ongoing hippocampal theta rhythm[6–8]. Inspired by these findings, we have developed a rhythmic sensory stimulation (RSS) approach to modulate episodic memory performance[9–11]. This approach enables the modulation of visual and auditory cortices following the application of RSS at theta frequency (4 Hz), with the two modalities being either in-phase or out-of-phase. This way, the sensory inputs that reach the hippocampus synchronously should be more likely to be associated, as compared to inputs arriving asynchronously. Indeed, previous studies by our and other labs found that associative memory performance was significantly better in the in-phase condition than in the out-of-phase condition[9,11,12].

This approach is promising to enhance episodic memory as it is non-invasive and inexpensive. Importantly, understanding the exact cognitive and neural mechanisms responsible for the RSS-induced memory effects may help to develop therapeutic interventions for clinical and healthy aging populations with impaired episodic memory. As an example, the RSS-induced memory effects might be challenged by trial-by-trial and/or) individual variability in entraining multisensory regions[11], further reaching the hippocampus. For example, the peak frequency of brain rhythms could differ between individuals or between cognitive states within an individual[13]. Furthermore, individual brain states fluctuate moment-by-moment, which determines how external stimuli are processed[14–17]. A recent study failed to show the RSS-induced memory effects, which raises the very important question on what the most effective RSS protocols are for enhancing episodic memory[18]. The mixed evidence on whether RSS can reach higher-level regions beyond sensory regions has also been discussed in recent investigations on Alzheimer's disease (AD) mouse models[19]. Several studies have shown that 40 Hz RSS can entrain both the sensory regions and the hippocampus. Moreover, AD pathology in those areas was significantly reduced. Hippocampal-related cognitive task performance, such as spatial memory, was also improved after the RSS[20,21]. In contrast, other studies have

[1]Department of Neuroscience, Physiology and Pharmacology, Division of Biosciences, University College London, London, UK. [2]School of Psychology and Neuroscience and Centre for Neurotechnology, University of Glasgow, Glasgow, UK. [3]School of Psychology and Centre for Human Brain Health, University of Birmingham, Birmingham, UK. ✉e-mail: danying.wang.09@ucl.ac.uk; Simon.Hanslmayr@glasgow.ac.uk

not found evidence for entrainment of hippocampal neurons at 40 Hz via visual 40 Hz RSS[22,23].

One variable that could explain the variance in finding RSS in downstream regions (such as the hippocampus) is the brain state during the stimulation[19]. Supporting this notion, Wang et al.[24], found that the gamma phase locking response to a 37.5 Hz audio-visual flickering stimulus was significantly higher for subsequently remembered compared to subsequently forgotten trials. It is also well-known that the amplitude of steady-state visual and auditory evoked potentials (SSVEPs and SSAEPs) is modulated by attention[25,26]. Therefore, variations in brain state could explain variations in finding evidence for RSS in downstream regions and, as a result, downstream effects on behaviour.

The current study investigates if the trial-by-trial variability in entrainment strength is modulated by brain state, and whether it can explain variations in the multisensory theta-induced memory effect (TIME) induced by the 4 Hz multisensory RSS. To do this, we recorded participants' brain activity using MEG while they underwent a standard version of the audio-visual RSS memory task (Fig. 1[9,11]). During the encoding phase, the luminance and the volume of neutral videos and sounds were modulated by a theta (4 Hz) sine wave with two phase offsets, 0° (in-phase) and 180° (out-of-phase). During encoding, participants were presented with audio-video pairs, and in a later recall task were cued with a sound and asked to recall the corresponding video. To study the modulation of brain state on the trial-by-trial variability in entrainment strength, we focused on the pre-stimulus alpha power. We hypothesised that pre-stimulus alpha power should predict whether a trial follows the RSS more faithfully.

## Methods
### Participants
29 young healthy participants participated in this experiment (12 male and 17 female; mean age: 24.3 years; range: 18–35 years). Participants' sex was determined by self-reports. Participants' race or ethnicity was not collected. All participants had normal or corrected-to-normal vision and normal hearing. Of the 29 participants, five were left-handed, one was

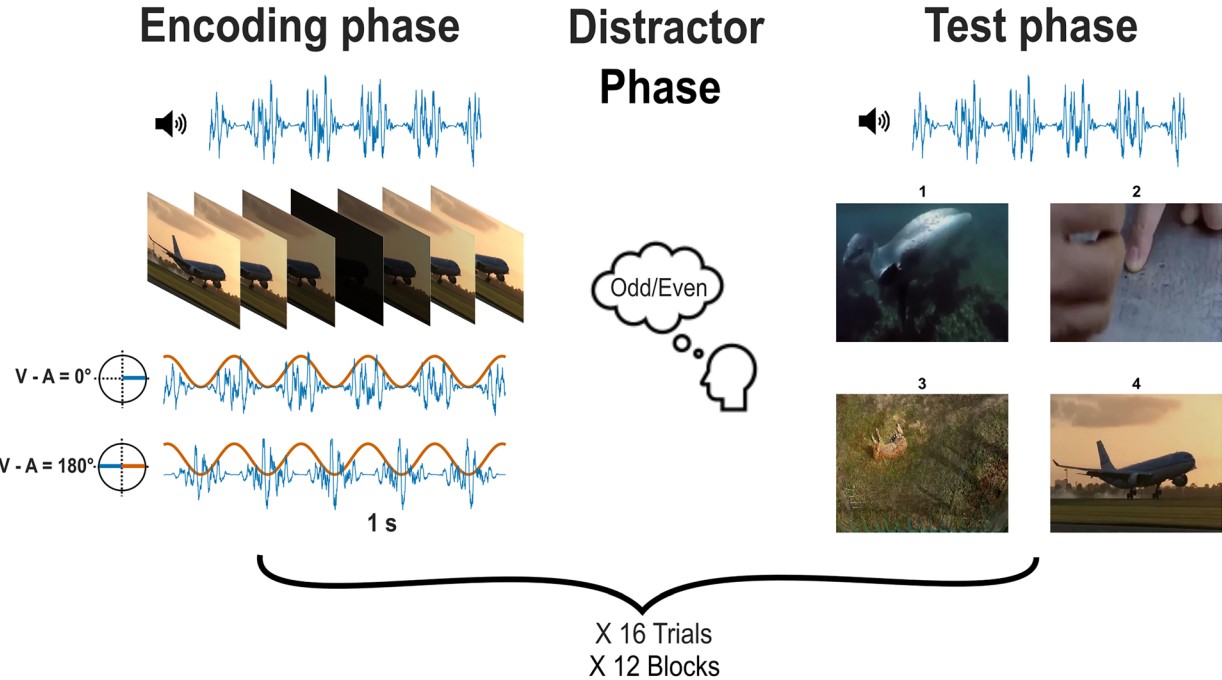

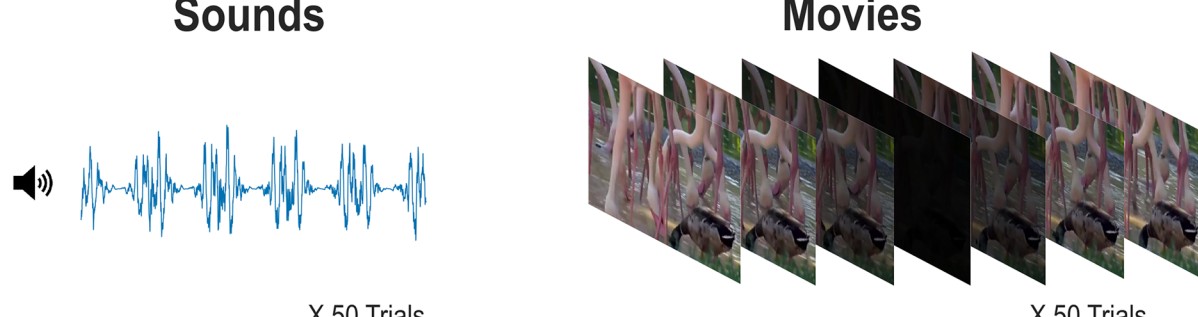

**Fig. 1 | Experiment procedure.** During the encoding phase, participants were presented simultaneously with 3 s flickering videos (orange) and 3 s fluttering sounds (blue) that were both modulated at 4 Hz and either in-phase (0° phase offset) or out-of-phase (180° phase offset). After 16 trials, participants were asked to judge whether a number presented on the screen was an odd number or an even number for 20 trials. Then, participants were asked to recall the paired movie when one of the fluttering sounds was presented. All 16 encoded trials were tested. After completing the 12 blocks of such memory tasks, participants were asked to listen to another 50 trials of fluttered sounds, then watch another 50 trials of flickered videos. All sounds and movies were modulated at 4 Hz. Participants brain activity was recorded using MEG. The images shown here were taken from the copyright-free videos downloaded from YouTube.com.

ambidextrous, and 23 were right-handed. Participants were recruited from the University of Glasgow School of Psychology and Neuroscience Subject Pool (https://participants.psy.gla.ac.uk/). All participants were paid £25 for their participation. The study has been approved by the Ethics Committee of the Faculty of Information and Mathematical Sciences at the University of Glasgow (FIMS 300200098). The behavioural data from four participants were excluded due to their poor, chance-level memory performance. The MEG data from five participant including the four participants whose behavioural performance was at chance-level, were excluded. The additional exclusion was because the participant did not complete the unimodal tasks. The behavioural data from 25 participants and the MEG data from 24 participants were retained for the final analysis. A power analysis using G*Power[27] was conducted based on the effect sizes averaged across four previous studies from our lab, Cohen's $d = 0.765$. Total sample size 25 is required, given alpha = 0.05 (one-tailed sample t-test) and 90% statistical power. This study was not pre-registered.

## Stimulus material

The visual and auditory stimuli were taken from the same stimulus set as those used in Wang et al.[24]. Movie clips of 3 s each had 91 frames in total with a frame rate of 30 frames/s, which was converted from the original 25 frames/s using HandBrake (https://handbrake.fr/). The movie clips consisted of documentaries on natural scenes, animals, architecture or human activities. 400 movie clips were modulated at 4 Hz with luminance changing between 0% and 100% (initially starting at 50% luminance). 400 sound clips were preprocessed using Audacity software (ver. 2.4.2 https://www.audacityteam.org/) as in Clouter et al.[9]. Each sound clip was presented concurrently with a movie for 3 s, with a lag of 40 ms, which compensated for the fact that auditory stimuli are processed faster than visual stimuli[24]. Sound amplitude was modulated at 4 Hz between 0 and 100% with a sine wave at 0° and 180° phase offsets relative to the modulation of the movie clips. The sound clips were not directly related to the contents of the videos. Each sound clip was taken from one of the eight sound categories as described in Wang et al.[24], with 36 sound clips for each category. The stimuli were allocated to three experimental tasks. 192 sounds and movies were used for the memory task. 56 sounds and movies were used for the unimodal task. The 56 sounds were modulated at only 0° phase offset from the modulation of the movies. 96 sounds and movies were used for the perceptual synchrony task. The 192 sounds and movies used for the memory task were randomly divided into two sets of equal size (96 sounds and 96 movies per set), with the constraint that the number of sounds for each sound category was equal. The same was done to the 96 sounds and movies used for the perceptual synchrony task. The assignment of the tasks and phase offset conditions to each sound and movie set was counterbalanced across participants.

The experiment was programmed with MATLAB (MathWorks) using the Psychophysics Toolbox extensions[28–30]. The visual stimuli were presented on a PROPixx projector with a refresh rate of 120 Hz. The presentation screen was ~115 cm from the participants' eyes. Auditory stimuli were presented with ER-30 tube-phone drivers (Etymotic Research) via Doc's Promold ear pieces to avoid electronic audio noise interference with the MEG recording. The physical presentation of phase offsets between movies and sounds, as well as the frequencies of the movies and sounds, was verified using the PROPixx projector Pixel Mode Triggers and a microphone placed at the end of the sound delivery tubes.

## Procedure

Participants provided informed consent before entering the MEG scanner. After they were given the task instructions, they practiced the procedure with four example trials. The formal experiment consisted of 12 blocks of an associative memory task, followed by two unimodal source localiser tasks and another 6 blocks of a synchrony judgement task. Each associative memory task block included an encoding phase, a distractor phase and a retrieval phase (Fig. 1). During the encoding phase, participants were presented with one of 16 movie clips paired with one of 16 sound clips. Each

trial started with a fixation cross with an inter-trial interval between 1 and 3 s. A sound-movie pair was presented for 3 s after the fixation cross. Participants were then instructed to make a judgment as to how well the sound suited the contents of the movie by pressing the keys on a Lumitouch response pad. The instruction screen was presented until a response was made. The ratings ranged from 1 (thumb key, the sound does not suit the movie at all), 2 (index key), 3 (middle key), 4 (ring key), to 5 (pinkie key, the sound suits the content of the movie very well). Participants were instructed to remember each sound-movie pair, as their memory of the association would be tested later. Within each block, four sounds from four sound categories were associated randomly with the 16 movies with the constraint that the number of sounds for each phase offset condition was equal in each sound category.

After the last trial of the encoding phase, participants were instructed to do the distractor task. A series of numbers was presented one by one, and participants were instructed to respond as fast and accurately as possible, indicating whether the number presented was an odd or an even number. They were asked to press the index finger key if the number was odd and press the middle finger key if the number was even. The distractor task included 20 trials, and participants were shown their accuracy, i.e. the proportion of correct trials of all 20 trials, at the end of the task.

Upon completion of the distractor task, participants were instructed to start the associative memory retrieval task. The retrieval phase consisted of 16 trials, during which all 16 sound-movie pairs presented during the encoding phase were tested. Each trial started with a fixation cross with an inter-trial interval between 1 and 3 s. Participants were presented with one of the 16 sounds heard during the encoding phase for 3 s, along with four still images from four of the movies from the encoding phase. Participants were instructed to select the movie that was presented with the sound in the encoding phase using the keys such that the index key corresponded to movie 1, the middle key corresponded to movie 2, the ring key corresponded to movie 3, and the pinkie key corresponded to movie 4. The screen of the four still images, along with the instructions, was presented until a response was made. All four movie options were associated with sounds from the same sound category presented during the encoding phase. Participants were allowed to take a break between blocks.

After the completion of the 12 associative memory task blocks, participants were given the instruction to finish the two unimodal localiser tasks. The unimodal task consisted of 50 trials of fluttered sound clips only, followed by 50 trials of flickered movie clips only. The sound and movie clips were different from the ones used in the associative memory task, to avoid memory components. On each trial, participants were asked to rate how pleasant the sound or movie was by using the keys that thumb key corresponded to 1 (the sound or the movie was very unpleasant), index key corresponded to 2, middle key corresponded to 3, ring key corresponded to 4, pinkie key corresponded to 5 (the sound or the movie was very pleasant).

Lastly, participants were instructed to perform a 6-block synchrony judgement task. The task was in the same format as the encoding phase. However, instead of memorising the sound-movie associations, participants were instructed to judge whether each sound-movie pair was presented in synchrony (in-phase, 0° phase offset) or asynchrony (out-of-phase, 180° phase offset). Participants were instructed to indicate asynchrony by pressing the index finger key and indicate synchrony by pressing the middle finger key. Each task block consisted of 16 trials. The 96 sound-movie pairs were different from the sounds and movies used in the encoding phase, to avoid the involvement of memory.

## MEG acquisition and preprocessing

Participants' brain activity was recorded with a 306-channel TRIUX neo MEG system from MEGIN (204 Gradiometers and 102 Magnetometers). An empty room recording was conducted for 3 min before each participant arrived at the lab. Participants' eye movements and ECG were recorded with two EEG electrodes placed above and below the right eye (vertical eye movements), two electrodes placed 1 cm to the left of the left eye and to the right of the right eye (horizontal eye movements), and two electrodes placed

close to the right and left clavicles (ECG). Participants' eye movements were also recorded using the EyeLink 1000 (SR Research) eye tracker that was positioned ~100 cm from participants' eyes. Calibration and validation were done by asking participants to follow a dot presented on the screen before starting recording. The online signals (MEG, EOG, ECG and Eye tracking positions) were sampled at 1000 Hz, with a highpass filter at 0.03 Hz, a lowpass filter at 330 Hz and continuous head position tracking (cHPI). All participants sit upright at a gantry angle of 60° inside of a magnetically shielded room.

Offline MEG data were preprocessed and analysed using the MNE-python[31], followed the FLUX pipeline[32] and the pipeline from Palva lab. Continuous brain and empty room data were high-pass filtered at 1 Hz and low-pass filtered at 148 Hz. Bad channels were marked, and head positions were computed. Next, MaxFilter with temporal signal-space separation (tSSS) was applied to interpolate the bad channels, transform different recording runs to the reference head position (the head position that was measured during the unimodal task, except one participant whose unimodal task was done in a separate day. The participant's multimodal data recording runs were referenced to the head position in the middle of the multimodal runs), and compensate for the signal interference due to head movement. Line noise at 50, 100 and 150 Hz was removed using a notch filter. Excessive muscle artefacts were identified before applying an independent component analysis (ICA). ICA components suggesting eye movement, heartbeat and muscle artefacts were removed from the data. After cutting data into epochs with 2 s pre-stimulus and 5 s post-stimulus and baseline correction (using data 1 s before stimulus onset), epochs with artefacts were manually rejected by visual inspection. Mean of percentage of rejected trials was 18.8% (range 0–56%) in the unimodal trials; mean of percentage of rejected trials was 18.0% (range 1.6–45.8%) in the multimodal trials.

## MRI acquisition and MEG source reconstruction

Structural magnetic resonance images (MRIs, T1-weighted) were acquired for each participant using a 3 T Trio Siemens scanner (3D MPRAGE, $1 \times 1 \times 1$ mm resolution). Anatomical reconstruction and construction of a single-layer boundary element model (BEM) were done using Freesurfer (https://surfer.nmr.mgh.harvard.edu/). The individual MRI was aligned to the participant's digitised scalp landmarks (nasion, left and right pre-auricular points) and scalp shape digitised before the MEG data recording. Volume source space was set up by a 5 mm grid covering the volume of the brain. The forward solution was constructed based on the volume source space in relation to the head position with respect to the MEG sensors.

The linearly constrained minimum variance (LCMV) beamformer was applied to the unimodal data to localise the unimodal visual and auditory 4 Hz steady state evoked activity. The data covariance matrix was estimated using the time window of 0.75–2.75 s after stimulus onset. The noise covariance was estimated using the time window of $-1$–0 s before stimulus onset for whitening to combine different channel types (magnetometers and gradiometers). The spatial filter was computed with reduced rank of the data (~70 due to the MaxFilter and ICA applications), hence no further regularisation was needed, and optimising the orientation of the sources. The resulting spatial filter was applied to the epochs to project the sensor data to the source space. Then the inter-trial coherence (ITC) was computed between 3.5 and 4.5 Hz at each source using Morlet wavelets with 7 cycles. The ITC averaged between 0.75 and 2.75 s was normalised by subtracting the ITC averaged between $-1$ and $-0.5$ s and dividing by the ITC averaged between $-1$ and $-0.5$ s. The normalised ITC at individual source space was morphed to the Freesurfer's 'fsaverage' T1 weighted MRI using the Symmetric Diffeomorphic Registration method[33]. Thus, the source ITC can be grand-averaged across participants. The coordinates for auditory and visual regions-of-interest (ROIs) were determined by where the maximum grand average ITC was shown on the fsaverage template MRI.

The multimodal source reconstruction followed the same steps as the unimodal source reconstruction except that the data and noise covariance were estimated using the data over left and right hemispheres separately, as suggested by Murzin et al.[34], to solve the issue that beamforming analysis has a poor performance in reconstructing highly correlated source activity, such as auditory stimulation. The spatial filter that was computed based on the left and right hemispheres was applied to the epochs separated by left and right hemispheres, respectively. After morphing the reconstructed time series to the fsaverage template MRI, data were extracted from the left auditory ROI, the right auditory ROI and the visual ROI, each defined based on the unimodal source reconstruction.

To solve the problem of ambiguous dipole orientation caused by the source reconstruction procedures, we manually forced the signs of source reconstructed time series to be consistent across hemispheres, regions and participants. We first computed the grand average unimodal source event-related fields (ERFs) and unimodal sensor ERFs. Since the left auditory source ERFs and the right auditory source ERFs were roughly in-phase, they were averaged to result in one unimodal auditory source ERF. We extracted the unimodal sensor ERFs from one gradiometer sensor and one magnetometer sensor, which showed the strongest evoked power averaged between 3.5 and 4.5 Hz and between 0.75 and 2.75 s. The grand average magnetometer auditory sensor ERFs were in-phase with the visual sensor ERFs. In contrast, the grand average gradiometer auditory sensor ERFs and the auditory source reconstructed ERFs were 180° phase shifted relative to the gradiometer visual sensor ERFs and the visual source reconstructed ERFs. Therefore, we used the magnetometer data as a reference to realign the source unimodal ERFs. The grand average unimodal visual source reconstructed ERFs were flipped 180° to be aligned with the magnetometer sensor ERFs, and the auditory source reconstructed ERFs remained the same. Then, each participant's left and right multimodal auditory source and visual source time series were band-pass filtered between 1.5 and 9 Hz before computing the ERFs, to be consistent with the previous study[11]. The ERFs were normalised by subtracting the mean of the whole time series and then divided by the standard deviation of the time series. The z-score transformed ERFs in each phase offset condition were plotted along with the z-score transformed grand average unimodal source ERFs. The signs of the multimodal time series were flipped by multiplying by $-1$ if the multimodal data in the 0° phase condition were not in-phase with the unimodal time series in the corresponding modality. The signs were applied to all trials in the corresponding modality, regardless of the phase offset condition. To confirm that the flipping procedure did not bias the results in favour of our hypothesis, we did a control analysis to compare the grand average ERFs within visual and auditory modalities between the 0° and 180° phase conditions. The phase difference between the 0° and 180° conditions in the auditory ROIs was 180° and the phase difference between the two phase offset conditions in the visual ROI was 0°, which remained constant before and after flipping.

## Bayesian analyses for testing the null hypothesis

Bayesian statistics was conducted to test the null results where the null results were interpreted. All Bayesian analyses were performed using JASP[35]. For the paired samples t-test, the default Cauchy prior of scale = 0.707 was chosen. For the repeated measures ANOVA, the default r scale = 0.5 was chosen for the coefficient prior of the fixed effects.

## Statistical analysis of MEG source reconstructed data

To compute the instantaneous phase differences on average level, the sign-flipped source reconstructed time series in the auditory ROIs were averaged. The individual ERFs at each ROI were grand averaged, baseline corrected and bandpass filtered between 1.5 and 9 Hz. The grand averaged ERFs were then Hilbert transformed, and phase angles were computed from the resulting analytic signal. The instantaneous phase differences were computed between the auditory and visual ROIs for each phase offset condition. The Rayleigh test and the V test were used to test circular uniformity of the instantaneous phase differences between 1 and 2 s after stimulus onset (to avoid influences of evoked responses at onset and offset) in each phase offset condition.

To measure the phase entrainment strength, we applied the Hilbert transformation to the bandpass filtered (1.5 and 9 Hz) single-trial time series at each ROI. Then the instantaneous phase differences were calculated between the auditory and visual Hilbert-transformed data for each phase offset condition. The single-trial phase entrainment measure was computed for each trial and each time point by calculating the resultant vector length of a vector that consisted of the data value at a time point and the theoretical phase offset (0° or 180°). The resultant vector length was averaged across time between 1 and 2 s, resulting in one value per trial. A 2 (phase offset condition) × 2 (subsequent memory performance) repeated-measures ANOVA was conducted on the averaged single-trial phase entrainment values for subsequently remembered and forgotten trials in each phase offset condition. The same pairwise comparisons were tested using paired sample t-tests, as done in the previous study[11], if a significant interaction was shown. Since the analyses were a replication of our previous study, all alternative hypotheses of the t-tests were one-sided. Test for normality indicated no deviation from normality. A linear mixed-effects model (LME) was built using RStudio and lme4 to account for the confounds of subjective ratings on how well a sound suited the content of a movie for a trial. The model was built using RStudio and the package lme4[36]. The model included the interaction between phase offset condition and subsequent memory performance, the interaction between phase offset condition and subjective rating and all lower-order terms, including a random effect (random slope) 'subjects' and control parameters of the Nelder Mead optimiser and removal of the derivative calculations to resolve the non-convergence of the model.

The same procedure was applied to the single-trial back-sorting analysis. Instantaneous phase differences were averaged across 1 and 2 s using scipy's circmean function (https://scipy.org/) to get one value per trial. Depending on the mean phase direction of a trial, we sorted the trials into two bins, centred at 0° and 180°, each with a bin width of ±90°. To reduce the trial number bias in each bin, we randomly selected a subset of trials with the minimum trial number—1 for each individual from each bin and calculated the proportion of remembered trials. This procedure was repeated 100 times, and the proportion of remembered trials in each bin for each repetition was averaged across the 100 repetitions. The same analysis was done for sorting the trials into four bins (Fig. S1A), centred at 0°, 90°, 180°, and 270°, each with a bin width of ±45°. Paired samples t-test was conducted to compare the statistical difference between phase bins.

Back sorting trials according to their actual phase differences might cause the distribution of movies and sounds to be not counterbalanced across phase offset conditions anymore. To account for this confound, we calculated how frequently a movie or a sound was remembered, as well as how frequently the movie or the sound was assigned to 0° phase bin, out of all 24 participants who were valid for the MEG analyses. Then we sort the movies or the sounds into the bin of low proportion that a movie or a sound was assigned to 0° bin, and the bin of high proportion that a movie or a sound was assigned to 0° (Fig. S2A, C). Independent samples t-test was used to compare the recall accuracy of the movies or the sounds between the bins. Scipy's normaltest function was used to test whether the distribution of the proportion when a movie or a sound was assigned to 0° bin was a normal distribution (Fig. S2B, D). To control for the memorability of movies, we randomly sub-selected equal numbers of the most memorable movies and the least memorable movies for 100 times. In each repetition, we calculated the recall accuracy in 0° phase bin and 180° phase bin and averaged the recall accuracy in each phase bin across all repetitions. A paired samples t-test was conducted to compare the statistical difference between phase bins based on the sub-selected trials. Test for normality indicated no deviation from normality for the data that was compared using t-tests. Further, a generalised linear mixed effects model (GLME) was built using. The model included fixed factors 'phase bin' and 'subjective rating', and random slopes 'subjects', 'movie ID' and 'sound ID', with control parameters of the optimx optimiser L-BFGS-B and removal of the derivative calculations to resolve the non-convergence of the model.

## Pre-stimulus oscillatory activity analysis

The time frequency representations of power were calculated using Morlet wavelets with 4 cycles to each epoch during the encoding task between 1 and 30 Hz at a step of 0.5 Hz. Pre-stimulus alpha power between 0.7 and 0.2 s before stimulus onset was calculated by averaging between 8 and 12 Hz. The pre-stimulus alpha power was averaged across trials in the matched condition, where the trials' actual phase differences between visual and auditory 4 Hz activity matched with the experimental conditions. The pre-stimulus alpha power was averaged across trials in the mismatched condition, where the trials' actual phase differences did not match with the condition labels. Mean pre-stimulus alpha power in each condition was baseline-corrected by the mean alpha power between post-stimulus 3.2 and 3.7 s to avoid stimulus offset evoked response as well as motor responses (mean RT of the valid trials was 1.46 s; median RT of the valid trials was 1.05 s; RT of 67% trials was longer than 0.7 s). Since the stimulus length was 3 s, on average, the activity after 3.2 s should not involve stimulus offset response. The difference between the mismatched pre-stimulus alpha power and the matched pre-stimulus alpha power was calculated for all gradiometer sensors and magnetometer sensors. It is well established that decreases in pre-stimulus alpha power are linked to attention and predict better subsequent task performance (see[37–39]). Therefore, a one-sample permutation t-test with spatio-temporal clustering was conducted with 1000 permutations and a right-tail at an alpha level of 0.05, to compare if the condition difference was significantly higher than 0 for gradiometer sensors and magnetometer sensors, respectively.

To test if increases in tonic alpha power drive the difference in pre-stimulus alpha power between conditions, we split the trials into the first half and the second half depending on the trials' temporal positions during the encoding task. The mean alpha power in each condition was averaged across gradiometer sensors and magnetometer sensors within the significant clusters. A 2 × 2 Repeated Measures ANOVA with the factors trial order (1st vs 2nd half) and whether the actual phase differences matched with the experimental phase condition (matched vs. mismatched) was conducted.

To test if the decrease in pre-stimulus alpha power was related to the task performance, within each half, trials were sorted into low and high alpha power bins depending on their baseline-corrected mean power averaged across 8 and 12 Hz and −0.7 and −0.2 s. In each alpha power bin, trials were sub-selected from each phase offset condition at equal sample size for 100 times. For each iteration, recall accuracy was calculated for each sub-selected phase offset condition in each alpha power bin, and in each half. A 2 × 2 × 2 Repeated Measures ANOVA with the factors trial order (1st vs 2nd half), phase offset condition (0° vs 180°) and pre-stimulus alpha power bin (low vs high) was conducted on the mean recall accuracy averaged across 100 iterations.

The analysis of eye tracking data followed the procedures in[40,41]. The raw eye tracking data recorded simultaneously with the MEG recordings were converted from voltage to pixel coordinates following the tutorial from the Fieldtrip (https://www.fieldtriptoolbox.org/getting_started/eyetracker/eyelink/#what-are-the-units-of-the-eye-tracker-data). Blinks were replaced by NaN if the values were smaller than −0.1 z-score of the time series data. The data was then epoched in the same way as was done to the MEG data. The gaze positions along the horizontal (x) and vertical (y) axes were binned into a 1000 × 1000 pixel grid. A 2D histogram was computed using the numpy module histogram2d. The histogram was then smoothed using a Gaussian filter from scipy's ndimage module with a smoothing factor of 5. A peak bin was found after averaging the 2D gaze heatmaps across 23 participants, excluding one participant whose eye-tracking data were not properly recorded. Time-resolved gaze density was computed for each trial and for each participant by computing the 2D histogram every 50 ms. The data along the vertical direction was averaged across 100 bins centred at the peak y bin, resulting in a data structure of time × horizontal position of the gaze density. Same one-sample permutation t-test was conducted with 1000 permutations with two-tail at alpha level 0.05 to compare the difference between the mismatch and match conditions. The time window was cropped at −0.7 and −0.2 s to be consistent with the pre-stimulus alpha

time window. The horizontal pixels were restricted to 100 bins centred at the peak x coordinate.

To source localise the pre-stimulus alpha power effect, the LCMV beamformer was applied to the multimodal data of gradiometer sensors. The data covariance matrix was estimated using the time window of 1–0 s before stimulus onset. The noise covariance was estimated using the empty room recordings. The data and noise covariance were computed with gradiometer sensors only. The spatial filter was computed with reduced rank of the data (the smallest rank between the data rank and the theoretical rank, that was ~66) and optimising the orientation of the sources. The resulting spatial filter was applied to the epochs to project the sensor data to the source space. The alpha power was computed the same way as done at the sensor level, using Morlet wavelets with 4 cycles, averaging between 8 and 12 Hz, −0.7 and −0.2 s, and baseline-corrected by subtracting and dividing by the mean alpha power between 3.2 and 3.7 s across trials in each condition. The baseline-corrected alpha power at individual source space was morphed to the Freesurfer's 'fsaverage' T1 template. The pre-stimulus alpha power difference between mismatch and match conditions was grand-averaged across participants and the grand-average alpha power difference was interpolated to the fsaverage MRI template.

We defined the region where the peak alpha power difference between mismatch and match conditions as the seed region. Each source reconstructed epoch data was morphed to the fsaverage MRI template first. The phase locking value[42] (PLV) was computed between the seed region and each other voxel by computing the complex time-frequency representations of the whole epoch using Morlet wavelets with 4 cycles. The resultant vector length of phase difference between the seed region and one of the rest voxels was computed across −0.7 and −0.2 s. The PLV was average between 3 and 8 Hz and normalised by the PLV across 3.2 and 3.7 s in each condition. A cluster-based one-sample permutation t-test was conducted with 1000 permutations and a right-tail at an alpha level of 0.05, masking the ROIs, including the left hippocampus, to compare if the difference in the normalised PLV between match and mismatch conditions was significantly higher than 0 in the hippocampus.

## Reporting summary

Further information on research design is available in the Nature Portfolio Reporting Summary linked to this article.

## Results

### Memory performance does not differ in the degree of cross-regional phase synchronisation averaged across trials

Our first step was to replicate the findings from the previous studies[9,11]. We used an LCMV beamforming method[43] to source localise the evoked power at 4 Hz in the unimodal conditions. Coordinates for the left and right auditory sources and one single visual source were determined by where the maximum power of 4 Hz was localised in the unimodal auditory condition and visual condition, respectively (Fig. 2A). After reconstructing the time series of multimodal data at source level using LCMV beamforming, the grand average ERF across 24 participants in each phase offset condition were extracted from the visual and auditory ROIs that were predetermined in the unimodal source localisation (Fig. 2C). The grand average ERFs were bandpass filtered between 1.5 and 9 Hz. After applying the Hilbert transformation to the bandpass filtered data, the instantaneous phase differences between the visual and auditory activity were computed for 1-s length (500 samples) starting from 1 s after stimulus onset (Fig. 2C). Rayleigh tests and V tests for the 0° and 180° conditions both lead to a rejection of the null hypothesis that the distribution of the phase differences was uniformly distributed, all $p$ values ≈ 0. The V tests further demonstrated that the distribution of the phase differences in each condition had a mean direction of its expected phase, 0° or 180°, respectively, both $p$ values ≈ 0.

The results suggest that the RSS effectively synchronised or desynchronised the corresponding sensory regions at 4 Hz. However, memory performance did not statistically differ between 0° and 180° conditions (Fig. 2B), $t(24) = −0.891$, $p = 0.809$ (one-sided), Cohen's $d = -0.178$, 95% CI

for Cohen's $d = [−0.508 \infty]$. The Bayesian paired samples t-test for the hypothesis that recall accuracy in the 0° condition was higher than in the 180° condition suggests strong evidence for the null hypothesis, $BF_{01} = 8.235$. In one of our previous studies[11], trial-by-trial variability of 4 Hz phase differences between visual and auditory activity did influence subsequent memory success, even though the phase difference on average was constant. Therefore, using the same analysis pipeline, we investigated if the level of single-trial phase synchronisation at 4 Hz could account for the subsequent memory success.

### Strength of single trial phase entrainment predicts subsequent memory performance

Following previous work from our lab[11], we used a single-trial phase entrainment measure (Fig. 3A), where the resultant vector length was computed for every time point between the observed phase difference (between visual and auditory sources, for example, 202° at the data point at 1.288 s in Fig. 3A) and the expected phase difference (for example, 180° in Fig. 3A). Therefore, a value close to 1 suggests high entrainment in that a given trial closely followed the RSS, whereas a value close to 0 suggests low entrainment, i.e. that the given trial deviated from the RSS. In the example in Fig. 3A, the entrainment strength of the single data point is 0.981, indicating strong entrainment to the 180° RSS. The entrainment measures of data points between 1 and 2 s after stimulus onset were averaged for each trial and then averaged over trials in each subsequent memory condition and phase offset condition. Replicating our previous findings[11], a 2 × 2 Repeated Measures ANOVA with the factors phase offset condition (0° vs 180°) and subsequent memory (hits vs misses) revealed a significant interaction between the two factors, $F(1, 23) = 10.613$, $p = 0.003$, Cohen's $d = 0.334$, 95% CI for Cohen's $d = [0.099, 0.570]$. Subsidiary paired-samples $t$ tests suggest that the interaction was a result of significantly stronger entrainment to 0° RSS in subsequently remembered trials than in forgotten trials in the 0° condition, $t(23) = 2.887$, $p = 0.004$, Cohen's $d = 0.589$, 95% CI for Cohen's $d = [0.219, \infty]$ (one-sided). In contrast, in the 180° condition, subsequently remembered trials showed significantly weaker entrainment than forgotten trials, $t(23) = −2.036$, $p = 0.027$, Cohen's $d = −0.416$, 95% CI for Cohen's $d = [-\infty, −0.061]$ (one-sided; Fig. 3C).

The results suggest that trial-by-trial variability in following the RSS may have obscured the memory effect when averaging across phase-offset conditions and across trials. Therefore, we back-sorted the trials based on their actual phase difference between auditory and visual cortices and re-labelled the trials based on their actual phase difference. To this end, we computed the instantaneous phase differences between band-pass filtered (1.5 and 9 Hz) visual and auditory activity, and averaged the phase differences between 1 and 2 s for each trial, again following the same procedure as in our previous study[11]. Each trial was sorted into one of the two phase bins (bin 1: $−\pi/2$ to $\pi/2$; bin 2: $\pi/2$ to $3\pi/2$) based on the averaged phase differences (Fig. 3B). The number of trials in each bin was equated by randomly sampling the one trial fewer than the minimum trial number between the two bins, from each bin. The proportion of remembered trials was calculated for each bin. This procedure was repeated for 100 times for each participant. The proportion of remembered trials calculated at each iteration was averaged across the 100 iterations. Paired samples t-tests revealed a significantly higher proportion of remembered trials in phase bin 1 than in phase bin 2, $t(23) = 3.504$, $p < 0.001$, Cohen's $d = 0.715$, 95% CI for Cohen's $d = [0.331, \infty]$ (one-sided, Fig. 3D). To be fully consistent with our previous approach[11], we also sorted the single trials into four bins centred at 0°, 90°, 180° and 270° based on the actual instantaneous phase differences (Fig. S1). We replicated the previous findings by showing that the mean recall accuracy in the 0° bin was significantly higher than in the 90°, 180°, and 270° bins, $t(23) = 2.196$, $p = 0.019$, Cohen's $d = 0.448$, 95% CI for Cohen's $d = [0.091, \infty]$; $t(23) = 2.052$, $p = 0.026$, Cohen's $d = 0.419$, 95% CI for Cohen's $d = [0.064, \infty]$; $t(23) = 2.600$, $p = 0.008$, Cohen's $d = 0.531$, 95% CI for Cohen's $d = [0.166, \infty]$, respectively (all one-sided, Fig. S1B). Whereas recall accuracy in the 90°, 180°, and 270° bins did not differ from each other statistically (90° vs 180°: $t(23) = −0.153$, $p = 0.880$, Cohen's $d = −0.031$, 95%

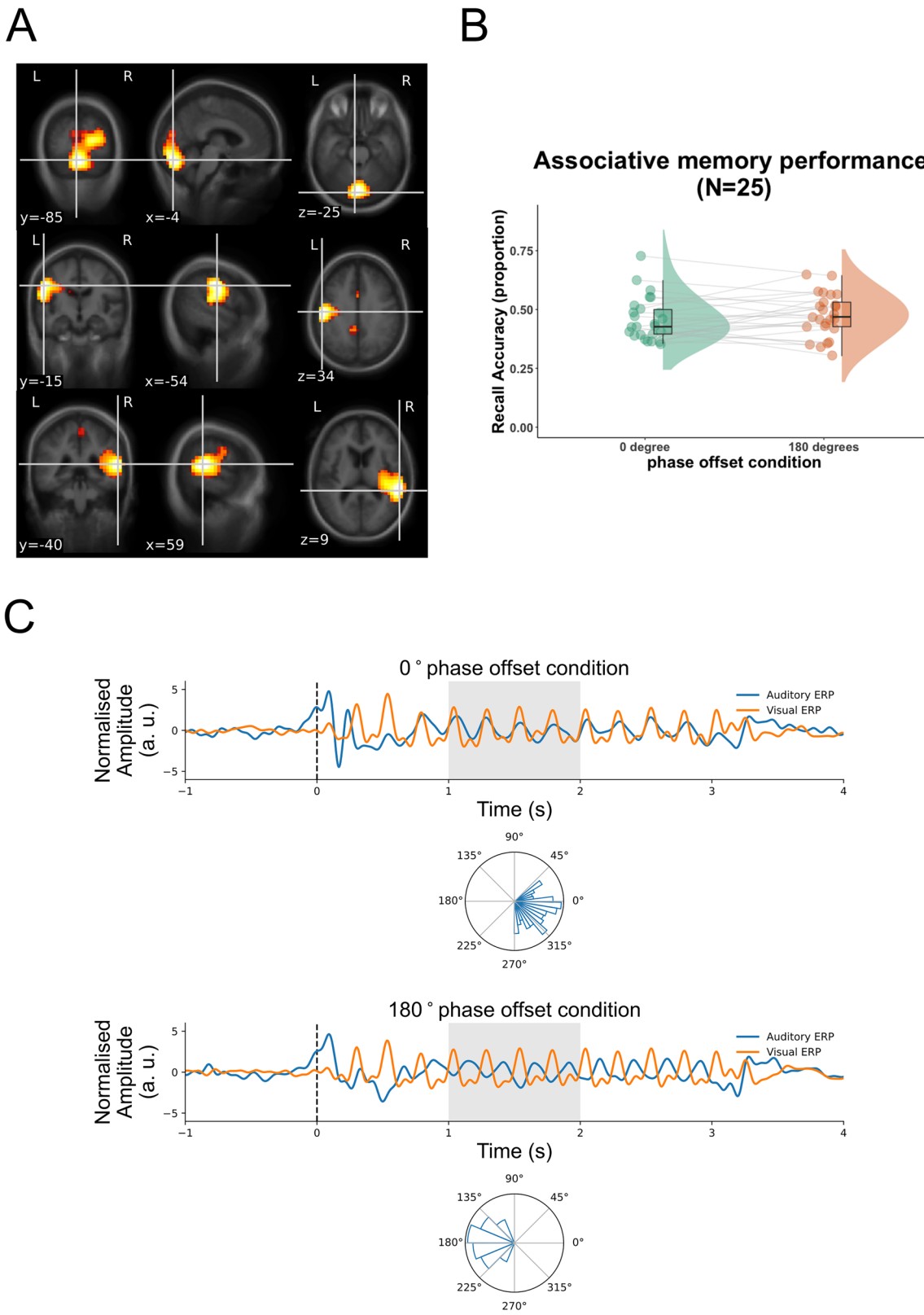

CI for Cohen's $d = [−0.431, 0.369]$; 90° vs 270°: $t(23) = 0.078$, $p = 0.939$, Cohen's $d = 0.016$, 95% CI for Cohen's $d = [−0.384, 0.416]$; 180° vs 270°: $t(23) = 0.230$, $p = 0.820$, Cohen's $d = 0.047$, 95% CI for Cohen's $d = [−0.354, 0.447]$).

Participants' subjective rating on how well a sound suited the contents of a movie influenced memory performance. A GLME of a fixed factor 'rating' and a random factor 'subjects' suggests that 'rating' significantly predicted memory performance, $\hat{\beta} = 0.133$, SE $= 0.028$, $z = 4.811$, $p < 0.001$, CI $= [0.076, 0.191]$. Therefore, to account for the confounds of subjective rating, LME was built on the entrainment measures with a fixed interaction term phase offset × memory performance and a fixed interaction term phase offset × rating, and a random effect of subjects. The interaction

**Fig. 2 | Memory performance and the degree of cross-regional phase synchronisation in each phase offset condition. A** source localisation of evoked 4 Hz power in the unimodal conditions. Top, visual source, MNI coordinates of ROI: −4, −85, −25. Middle, left auditory source, MNI coordinates of ROI: −54, −15, 34. Bottom, right auditory source, MNI coordinates of ROI: 59, −40, 9. Evoked power at source space was averaged between 3.5 and 4.5 Hz and between 0.75 and 2.75 s in the unimodal visual and auditory conditions. The values were normalised by the averaged evoked power of 0.5 s pre-stimulus baseline starting from 1 s before stimulus onset. **B** Associative memory task performance. Proportion of the correctly selected movie scenes that were associated with presented sounds in each phase offset condition. Individual data for each condition is shown in dots connected by grey lines (25 participants, including one who did not complete the unimodal conditions). **C** Phase differences between auditory and visual sources in each phase offset condition. In each phase offset condition, amplitude-normalised grand average ERF signals at the auditory source are in blue. Signals at the visual source are in red. Circular histograms of instantaneous phase differences between visual and auditory sources, and between 1 and 2 s (shaded time window on the ERFs) are plotted on unit circles, in 16 equally-sized bins.

between phase offset and memory was still significant, $\hat{\beta} = 0.019$, SE = 0.007, df = 23.27, $t = 2.739$, $p = 0.012$, CI = [0.005, 0.033]; whereas the interaction between phase offset and rating was not statistically significant, $\hat{\beta} = 0.0004$, SE = 0.003, df = 23.07, $t = 0.109$, $p = 0.914$, CI = [−0.006, 0.007]. Specifically, the strong entrainment to 0° RSS but weaker entrainment to 180° RSS led to subsequent memory success, while subsequently missed trials showed the opposite pattern, replicating our previous findings[11].

The results suggest that although the RSS did not always entrain a trial, when the trial followed the optimal (i.e. 0 phase lag) 4 Hz phase differences, memory was better than when the trial did not follow the optimal phase, which replicates our previous findings[11]. However, it is important to consider that when sorting the single trials into the 0° or the 180° bins, the distribution of stimuli (i.e. movies and sounds) is not counterbalanced across phase offset conditions anymore. This could lead to a potential confound between memory advantage in 0° and a greater proportion of memorable items. To address this concern, we conducted a number of control analyses, which are reported in Fig. S2. The control analyses suggest that the proportion of movies assigned to the 0° bin was normally distributed. Further, the memory advantage found in 0° was still significant even after randomly sampling equal amounts of trials from better remembered items and less remembered items for the 0° and 180° bins. A GLME was built with fixed factors 'phase bin' and 'subjective rating', and random factors 'subjects', 'movie ID' and 'sound ID'. The effect of 'phase bin' was still significant after taking the effect of stimuli into account, $\hat{\beta} = 0.238$, SE = 0.075, $z = 3.175$, $p = 0.002$, CI = [0.093, 0.387].

## Highly-entrained trials were preceded by decreased alpha power

After replicating our previous finding showing that trial-by-trial variations in entrainment strength explain subsequent memory performance, we interrogated whether this trial-by-trial variability in entrainment strength can be explained by variations in brain state. Specifically, we computed alpha power, which has been linked to attentional brain states for processing external information[44–49], between 0.7 and 0.2 s prior to stimulus onset and between 8 and 12 Hz. We then compared pre-stimulus alpha power for trials whose actual phase differences between visual and auditory 4 Hz activity matched with their expected phase difference (i.e. the experimental condition), with those trials whose phase differences did not match their expected phase difference. A cluster-based permutation one-sample t-test on all gradiometers and magnetometers showed that alpha power significantly decreased before the trials whose actual phase differences matched with their intended phase difference, as compared to the trials whose actual phase differences did not match with the intended phase difference (Fig. 4A, gradiometer sensor cluster $t = 76.614$; magnetometer sensor cluster $t = 69.936$; the effect size for the averaged alpha power over the cluster was: gradiometer sensor cluster Cohen's $d = 0.53$; magnetometer sensor cluster Cohen's $d = 0.52$). This analysis suggests that pre-stimulus brain state determines in part if a given trial accurately follows RSS. However, this relation between matching and unmatching trials and alpha power could also be driven by a hidden third variable, for instance, an increase in tonic alpha power over time, together with decreased reactivity to the RSS[50]. To test for this possibility, we averaged the pre-stimulus alpha power across the gradiometer and magnetometer sensors in the clusters that showed a significant difference between matched and mismatched trials and split the trials into the first and second half of the memory task for each participant. A

2 ×2 Repeated Measures ANOVA with the factors trial order (1st vs 2nd) and match between the actual phase bin and the experimental phase condition (match vs mismatch) suggested main effects of trial order and label match. The pre-stimulus alpha power in the second half of the trials increased significantly as compared to the alpha power in the first half of the trials, $F(1, 23) = 10.989$, $p = 0.003$, Cohen's $d = −0.362$, 95% CI for Cohen's $d = [−0.614, −0.111]$. The main effect of label match suggests that decreasing pre-stimulus alpha power effect before the matched trials was not simply due to the tonic alpha power changes over time, which remained significant after taking trial order into account, $F(1, 23) = 5.938$, $p = 0.023$, Cohen's $d = −0.165$, 95% CI for Cohen's $d = [−0.313, −0.016]$. Moreover, recent studies have linked fluctuations in alpha power to fixation-related eye movements[40,41]. Therefore, we have compared the fixation density between the mismatch and match conditions in the same pre-stimulus time window and showed that there was no significant difference (Fig. S4).

The results suggest that decreases in pre-stimulus alpha power predict how accurately a given trial follows the RSS. Next, we investigated if pre-stimulus alpha power also links with subsequent memory behaviour. As above, we split the trials into the first half and the second half depending on the trial order to address the potential confound of alpha power systematically changing across the time on task. To avoid conflating the variance, within each half of trials, we median split the trials into low pre-stimulus alpha power trials and high pre-stimulus alpha power trials according to their pre-stimulus alpha power that was averaged between −0.7 and −0.2 s, 8 and 12 Hz, and across all the gradiometers and magnetometers in the significant clusters. Then, for each pre-stimulus alpha power bin, we sub-sampled the same number of trials from 0° and 180° phase offset conditions, respectively. Mean recall accuracy was calculated for each phase offset condition. A 2x2x2 Repeated Measures ANOVA with the factors trial order (first vs second), phase offset condition (0° vs 180°) and pre-stimulus alpha power (low vs high) revealed a significant interaction between phase offset condition and pre-stimulus alpha power, $F(1, 23) = 5.448$, $p = 0.029$, Cohen's $d = 0.254$, 95% CI for Cohen's $d = [0.016, 0.493]$ (Fig. 4C). The interaction was driven by significantly higher recall accuracy in the low alpha power bin than in the high alpha power bin in the 0° phase condition, $t(23) = 2.747$, $p = 0.011$ (Bonferroni corrected), Cohen's $d = 0.433$, 95% CI for Cohen's $d = [0.088, 0.778]$; no significant difference in recall accuracy between low and high alpha power bins in the 180° phase condition was observed, $t(23) = −0.511$, $p = 0.614$ (Bonferroni corrected), Cohen's $d = −0.083$, 95% CI for Cohen's $d = [−0.411, 0.246]$. Bayesian Repeated Measures ANOVA suggests moderate evidence for the null hypothesis that there was no difference in recall accuracy between low and high alpha power bins in the 180° phase condition, $BF_{01} = 3.345$.

Next, we used an LCMV beamforming method[43] to source localise the significant difference in pre-stimulus alpha power between mismatch and match conditions that was exhibited at sensor space. After reconstructing the time series of the data at the source level using LCMV beamforming, alpha power (8–12 Hz) was computed for mismatch and match conditions using a Morlet wavelet transformation. The mean power between −0.7 and -0.2 s was normalised by the mean power between 3.2 and 3.7 s within each condition. The difference in normalised pre-stimulus power between mismatch and match conditions was interpolated to a standard MRI image using volumetric morphing (Fig. 4B). The strongest pre-stimulus power difference was localised to the posterior parietal cortex (PPC).

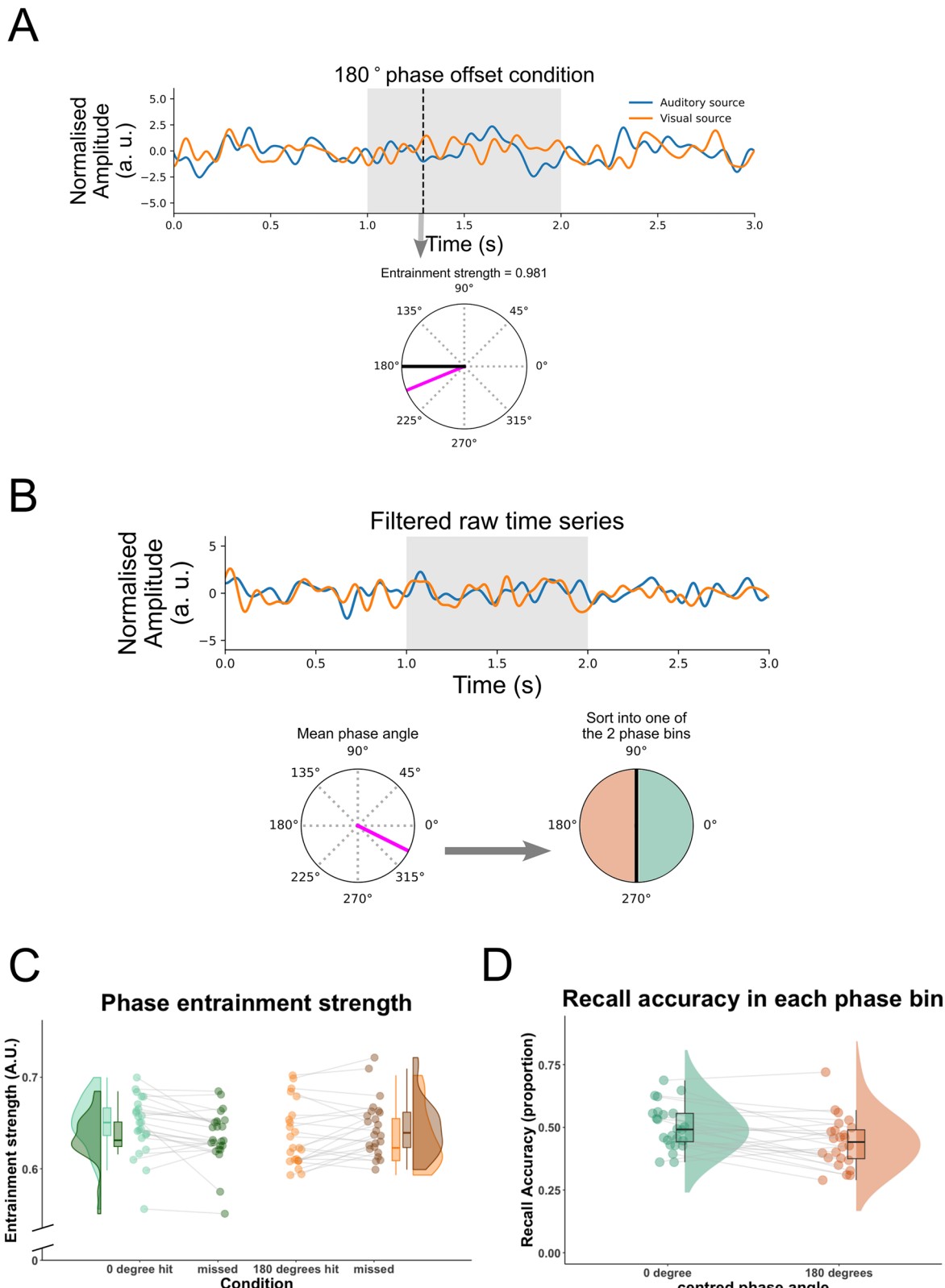

Finally, we investigated whether inter-regional coupling in 4 Hz was modulated by the match/mismatch condition. To this end, we calculated inter-regional phase coupling[42] between the PPC and all other regions. The results show that functional connectivity at theta frequency was significantly greater between the PPC and the hippocampus in the same pre-stimulus time window in the match compared to the mismatch condition (Fig. S3).

## Discussion

TIME reveals that episodic memory can be improved by synchronising theta activity in the auditory and visual cortices that are entrained by 4 Hz in-phase multisensory RSS[9]. TIME has been suggested to be influenced by trial-by-trial fluctuations in the entrainment strength of the RSS[11]. Here, we show that pre-stimulus alpha power modulates the trial-by-trial variability

**Fig. 3 | Trial-by-trial variability in 4 Hz phase differences between auditory and visual sources. A** Bandpass filtered (1.5–9 Hz) single trial from auditory (blue) and visual (red) sources in the 180° phase offset condition. The instantaneous phase difference at 4 Hz between auditory and visual sources at the time point (1.288 s) indicated by the black dashed vertical line is plotted on a unit circle (magenta line). Single-trial phase entrainment measure was calculated as the resultant vector length between the observed phase difference (magenta line) and the expected phase offset 180° (black line), which was 0.981 in this case. Then all the resultant vectors for every time point between 1 and 2 s (shaded time window on the single trial time series) were averaged for each trial. **B** Another band pass filtered single trial time series. The instantaneous phase difference for each time point between 1 and 2 s (shaded time window) was averaged and plotted on a unit circle (magenta line indicates the mean phase angle). Then each trial was sorted into one of the two phase bins (right) depending on the mean phase angle. **C** Single-trial phase entrainment measure was plotted as a function of subsequent memory (hit, green; missed, red) in each phase offset condition. **D** The proportion of remembered trials was calculated for the trials of one trial fewer than the number of minimum trial number between the two bins that were randomly sampled from each phase bin for each iteration. The recall accuracy was averaged across 100 iterations. In both **C** and **D**, individual data for each condition is shown in dots connected by grey lines (24 participants).

in entrainment strength, thus influencing subsequent memory performance. We successfully replicated the previous findings on TIME at a single-trial level, showing that single-trial phase synchronisation at 4 Hz improves the likelihood of memory success and that this is not due to an item effect. However, RSS did not always entrain the sensory regions at desired phase offsets, resulting in approximately half of trials not matching with their experimental conditions. Arguably, this led to a failure in replicating the TIME when averaged across trials. Pre-stimulus alpha power was increased for trials whose actual phase difference between auditory and visual 4 Hz activity did not match with the expected phase offset. Moreover, pre-stimulus alpha power modulated subsequent memory performance. The source localisation of the pre-stimulus alpha power between mismatch and match conditions points to a brain region that is related to attention, the PPC[51]. The findings enhanced our understanding on the RSS induced TIME and might provide insights on why recent studies could not replicate the TIME[18,52].

Despite the differences in experimental procedures and stimuli used in those experiments, our results suggest that the robustness of TIME very much depends on the trial-by-trial variability in entrainment strength. Serin et al.[18], and the current study both performed the experiment in the MEG suite. The auditory stimuli's quality might be influenced by the MEG-compatible auditory deliver system because auditory stimuli were delivered through long, thin silicone tubes into the magnetically shielded room to prevent interference with the MEG signals, which diminished the quality (i.e. reduced frequency bandwidths) of the sounds. Such changes likely cause variability in entraining the sensory regions, especially the auditory region, resulting in variable phase shifts compared to the visual stimuli. Our previous studies suggest that even though phase in sensory regions is entrained to the multisensory RSS on average, subsequent memory performance is better predicted by the actual phase difference on a single-trial level[11,24]. One of the procedural changes Serin et al.[18], made in their study was to block the synchronous and asynchronous trials, instead of intermixing them. Although they have not conducted the single-trial analysis, it would be interesting to see if blocking the trials can also introduce variability in entrainment strength, thus enabling to understand to what extent this could explain the lack of TIME on average level. In the current study, we successfully replicated previous findings from our group[11] showing that the single-trial entrainment strength in the 0° condition was positively related to subsequent recall, whereas in the 180° condition, stronger entrainment strength was related to subsequent forgetting. Additionally, sorting trials according to their actual phase difference results in significant TIME and replicates the 'hockey-stick' pattern, which refers to increased memory for the 0° condition compared to the 90°, 180° and 270° conditions[9,11]. Notably, this hockey-stick pattern points to an interaction between theta-phase-dependent and spike-timing-dependent plasticity as pointed out in a recent computational modelling study[24]. We further demonstrated that the memory effect after sorting trials based on their actual phase offsets was not driven by a bias towards more memorable sounds or videos being more likely to be sorted into 0° phase bin. Evidence from an intracranial EEG study has shown that the human Medial Temporal Lobe responds to theta (5.5 Hz) audio-visual RSS[53]. Therefore, the single-trial TIME is more likely to be supported by the interaction between the two synaptic plasticity mechanisms in the hippocampus. This interpretation would also

be in line with a recent MEG study that showed that hippocampal theta power is enhanced for synchronous vs. asynchronous multi-sensory speech stimuli[54].

Importantly, and going significantly beyond previous studies, our results show that pre-stimulus alpha power was decreased for trials whose actual audio-visual phase offset matched with the expected phase, as compared to the trials which did not match with the expected phase offset. A plethora of studies showed anticipatory alpha power decreases in tasks where attention was oriented spatially or temporally, thus predicting performance and reaction times of visual targets[55–58]. In a memory inhibition study, pre-stimulus alpha power has been found to increase before the to-be-inhibited items, as compared to the to-be-remembered items and the difference in the pre-stimulus alpha power was localised in the parietal cortex[59]. Moreover, the pre-stimulus alpha power decrease in the PPC has been linked with subsequent memory success regardless of whether the upcoming item was to-be-remembered or not-to-be-remembered. A proposed mechanism of the PPC alpha oscillations is to gate information flow, which increases engagement of the contralateral dorsal attention network associated with a target, and reduces information flow in the ipsilateral side of the distractor[60]. Such a gating mechanism could facilitate resource allocation in the cortical areas[61]. This is also consistent with findings in episodic memory formation and retrieval, which suggest that alpha power decreases reflect richer representation of information[62,63]. Therefore, the stronger theta phase coupling between the PPC and the hippocampus shown in the current study could reflect that the sensory information is more likely to reach the hippocampus.

Another line of research demonstrated that phase-locked responses (or entrainment) to rhythmic flickering stimuli are strongly modulated by attention, such that the brain follows a flickering stimulus more faithfully during high compared to low attentional states[25]. Modulation of oscillatory activity using auditory stimulation has been suggested to be highly state-dependent, such as closed-loop auditory stimulation during sleep[64]. Moreover, neuronal activity in the auditory cortex follows the auditory stimulation more faithfully in the desynchronised state. Whereas in the synchronised state, when low-frequency power is prevalent, the neuronal activity misaligns from the stimulus[65]. Our finding of decreases in alpha power preceding phase-offset-matched trials is consistent with these studies. Previous studies show that the state of phase entrainment, especially in the auditory domain, alternates with the state of alpha at a rate of approximately 0.06 Hz[66]. When alpha power is strong, auditory entrainment is weak and decoupled from the external stimulation. Whereas when alpha power is weak, auditory rhythmic activity shows stronger phase entrainment to the external stimuli. In light of these findings, our results suggest that fluctuation in trial-by-trial variability in entrainment strength is modulated by pre-stimulus alpha power. Pre-stimulus alpha power decreases for phase-offset-matched trials might reflect more engagement of attentional resources, which in turn increases the phase alignment to the external flickering stimuli.

Relatedly, we also show that pre-stimulus alpha power is linked with subsequent memory performance but only in the 0° condition. Low pre-stimulus alpha power resulted in a higher proportion of recalled trials compared to high pre-stimulus alpha power. However, for the 180° condition, recall accuracy did not differ significantly between low and high

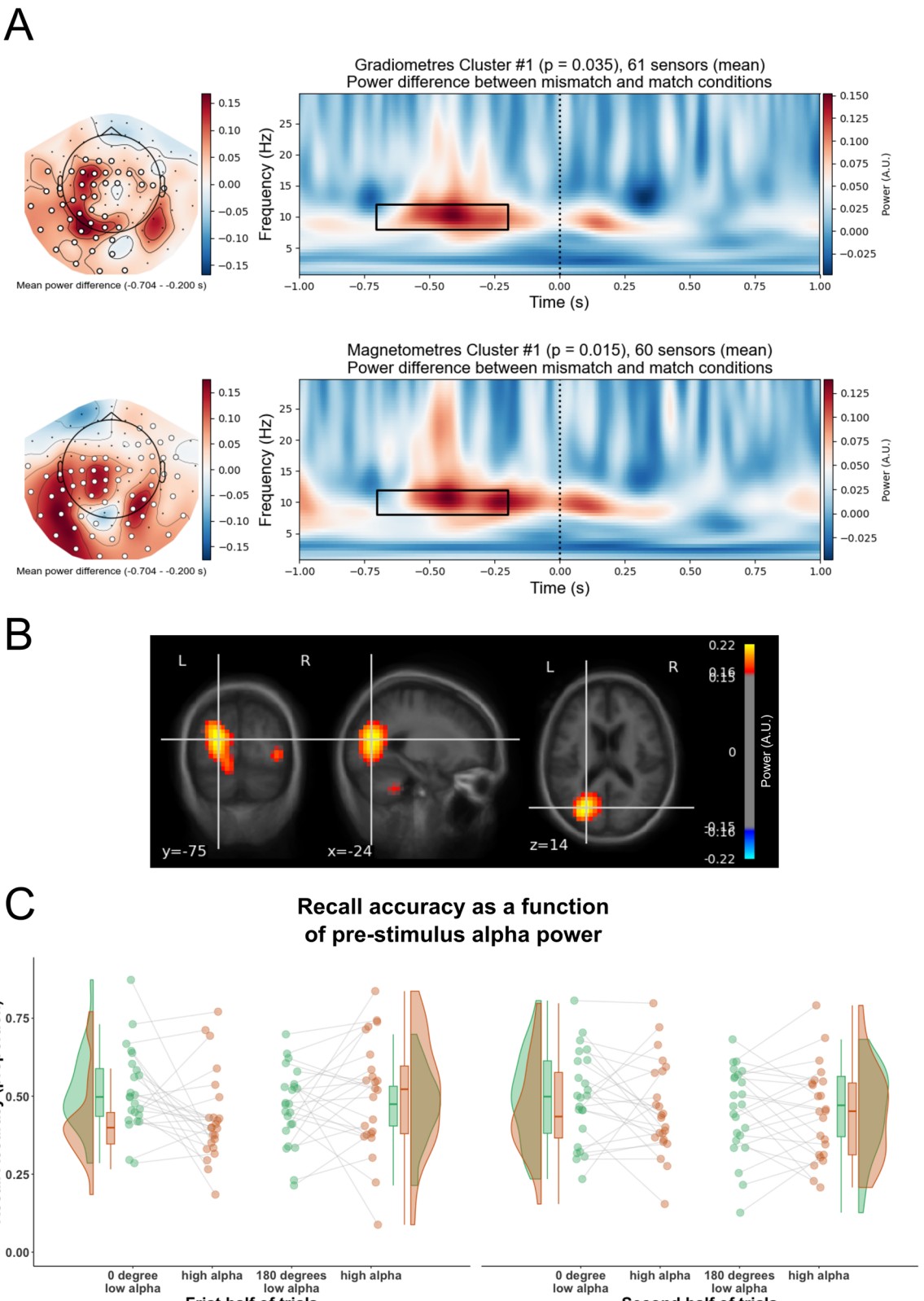

alpha power. This pattern of results could be due to two forces, attention and phase-offset pulling in the same direction for the 0° condition, but in opposite directions for the 180° condition. For the former, enhanced attention would both benefit memory encoding, because enhanced attention boosts memory[67], and it would also increase the phase-alignment with a condition that is known to be beneficial for memory. However, for the 180°

condition, enhanced attention would increase memory encoding, whereas it would lead to enhanced phase-alignment in a condition that is known to decrease memory encoding. Therefore, one could argue that the beneficial effects of attention, and the detrimental effects of phase-alignment cancelled each other out for the 180° condition, whereas the beneficial effects for memory were amplified in the 0° condition.

**Fig. 4 | Difference between mismatch and match conditions in pre-stimulus alpha power. A** Significant increase in alpha power prior to the trials whose actual phase differences did not match the intended phase differences (mismatch), as compared to the trials whose actual phase matched with the intended phase (match). Right, time frequency representations (TFRs) of grand average power difference between mismatch and match conditions, between −1 and 1 s, relative to the mean power of baseline 3.2 and 3.7 s across trials within a condition. Top, the power difference was averaged across 61 gradiometer sensors in the significant cluster. The vertical black dashed line indicates stimulus onset. Bottom, same as Top, but the power difference was averaged across 60 magnetometer sensors in the significant cluster. Left, topographic map showed the alpha power (averaged between 8 and 12 Hz, and −0.7 and −0.2 s, highlighted in the black rectangle in the right TFRs) difference between match and mismatch conditions for all gradiometer sensors (Top), and all magnetometer sensors (Bottom). The highlighted sensors indicate the sensors in the significant cluster. **B** Source localisation of the difference in pre-stimulus alpha power between mismatch and match conditions. Oscillatory power at source space was averaged between 8 and 12 Hz and between −0.7 and −0.2 s in each condition and was normalised by the averaged power of 0.5 s baseline starting from 0.2 s after stimulus offset. The difference in the normalised alpha power between the mismatch and match conditions was overlaid onto a standard MRI image in the MNI space. MNI coordinates of the strongest alpha power differences: −24, −75, 14. **C** Recall accuracy was plotted as a function of pre-stimulus alpha power in each phase offset condition, in each half of trials. In each half of trials, trials in each phase offset condition were median split into low alpha power bin (green) and high alpha power bin (red) according to the pre-stimulus alpha power averaged across −0.7 and −0.2 s and averaged across gradiometer and magnetometer sensors in the significant clusters, relative to the mean alpha power between 3.2 and 3.7 s over trials in match and mismatch conditions. The proportion of remembered trials was calculated in the low and high alpha power bins in each phase offset condition. Individual data for each condition is shown in dots connected by grey lines (24 participants).

Alpha oscillations could reflect internal processing modes that can be activated by attentional drift due to time on task, hence resulting in fatigue[50,68]. We replicated these findings in showing that pre-stimulus alpha power indeed increased across time during the task. However, after controlling for this alpha power drift over time, we still observed that trial-by-trial variability in entrainment strength is modulated by pre-stimulus alpha power. Additionally, the interaction between pre-stimulus alpha power and phase offset condition on subsequent recall accuracy was still significant after taking time on task into account. Perhaps future studies could investigate if the entrainment mode induced by the current RSS protocol is alternated with the alpha mode at 0.06 Hz as suggested by the previous study by Lakatos et al.[66]. This could help to optimise the RSS stimulation to be more effective in inducing the TIME. Alternatively, recent studies have shown that alpha power increases when fixation is maintained, whereas initiating eye movements to explore stimuli leads to a decrease in alpha power[40,41]. The difference in alpha power between the two experimental conditions could be the difference in visual exploration due to the complexity of the stimuli. No statistical evidence suggests that this could explain the pre-stimulus alpha power effect between mismatched and matched trials in the current study. The fixation density did not differ significantly between conditions in the same pre-stimulus time window where the alpha power difference was found.

In conclusion, our study provides insights into the variability and effectivity on using RSS to modulate memory performance. Our results demonstrate that variations in pre-stimulus brain state are the reason for why the brain does not always follow the RSS. Our findings might be also insightful for understanding the discrepancy in evidence on whether RSS can reach higher-level brain regions in rodents[23]. Having animals pay attention to the RSS is important for the rhythmic sensory inputs to reach the higher-level brain regions, such as the hippocampus. We show that functional activity at theta frequency was significantly greater between the PPC, where the strongest pre-stimulus alpha power difference was localised, and the hippocampus in the same pre-stimulus time window before the matched trials than before the mismatched trials. This supports that attentional state amplifies the effect of RSS reaching the downstream regions. Previous studies suggest TIME on the average level[9,11], which the current study and Serin et al.[18], did not replicate. Instead, the current study indicates that the TIME might be state-dependent. Previous studies have also suggested that the effect of brain stimulation on perception or visual short-term memory is state-dependent[69,70]. Here, we show that the effect of RSS on episodic memory is also state-dependent. This highlights the need for brain-state dependent stimulation protocols which individualise stimulation parameters, before we can confidently use RSS as a non-invasive therapeutic intervention in clinical populations with impaired hippocampal function.

## Limitations

The sample size in the current study was determined based on the effect sizes averaged across previous studies from our lab. We note that if the studies from Serin et al.[18], were included, it would require a total sample size of 54, given alpha = 0.05 (one-tailed paired sample t-test) and 90% statistical power. Although Serin et al.[18], was not published while we conducted the current study, a highly powered, pre-registered direct replication would be valuable for clarifying the robustness of the TIME. However, the current study contributes important insights into the conditions under which the effect may or may not emerge, and highlights the need to consider physiological variability in entrainment when interpreting null results.

## Data availability

Preprocessed data are available from https://doi.org/10.6084/m9.figshare.29468597. Any additional information required to reanalyse the data reported in this paper is available upon request.

## Code availability

All original code has been deposited at https://osf.io/skm4q/.

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

## Acknowledgements

This research was supported by grants from the Economic Social Sciences Research Council (https://esrc.ukri.org/, ES/R010072/1 to S.H. and K.L.S.) and the Medical Research Council Doctoral Training Program in Precision Medicine (MR/W006804/1 to E.M.). The funders had no role in study design, data collection and analysis, decision to publish or preparation of the manuscript. The authors would like to thank Gabriela Cruz, Máté Gyurkovics, Hamed Haque, and Felix Siebenhühner from the Palva lab for their help on the MEG preprocessing pipeline, and Frances Crabbe, Xuan Cui, Jacqueline McDiarmid, and Gavin Paterson for their help on MEG and MRI data acquisition, and everyone from the Neurotechnology, Cognition and Oscillations Lab and Professor Maria Wimber's lab at the University of Glasgow for their helpful inputs. The authors would also like to thank Tzvetan Popov for his suggestions and for sharing code on the eye tracking data analysis.

## Author contributions

Conceptualization, S.H., K.L.S. and D.W.; investigation, D.W. and E. M.; formal analysis, and writing—original draft, D.W.; writing—review and editing, D.W., E.M., K.L.S. and S.H.; funding acquisition, and supervision, S.H. and K.L.S.

## Competing interests

S.H. acts as scientific adviser to Clarity Technologies Inc. All other authors declare no competing interests.
