## [Transparent Peer Review file · Communications Psychology]

Pre-stimulus alpha power modulates trial-by-trial variability in theta rhythmic multisensory entrainment strength and theta-induced memory effect

Corresponding Author: Dr Danying Wang

Version 0:

Decision Letter:

Dear Dr Wang,

Thank you for your patience during the peer-review process. Your manuscript titled "Pre-stimulus alpha power modulates trial-by-trial variability in theta rhythmic multisensory entrainment strength and theta-induced memory effect" has now been seen by 3 reviewers, and I include their comments at the end of this message. They find your work of interest but raised some important points. We are interested in the possibility of publishing your study in Communications Psychology, but would like to consider your responses to these concerns and assess a revised manuscript before we make a final decision on publication.

We therefore invite you to revise and resubmit your manuscript, along with a point-by-point response to the reviewers. Please highlight all changes in the manuscript text file.

Editorially, we consider it crucial that the theoretical and methodological concerns, such as the potentially insufficient power to provide compelling evidence to support the central claim of the study and the alternative explanations of the in/out-of-phase analysis, are thoroughly addressed in the revised manuscript with additional analyses.

We also ask you to add a sensitivity power analysis to your manuscript. Please do not conduct a post-hoc power analysis based on the observed effect size in your study (cf. Lakens, 2022, <https://doi.org/10.1525/collabra.33267>).

Please ensure you follow our statistical guidelines when reporting statistics (<https://www.nature.com/commspsychol/submit/submission-guidelines#statistical-guidelines>). Please note in particular our requirements for the reporting and interpretation of null-results. Non-significant findings derived from null-hypotheses significance tests should be reported in full, but may not be interpreted. Where you interpret null results, this interpretation must be based on Bayes Factors or equivalence tests.

I am attaching an Editorial Requests Table that details critical reporting requirements for the revised manuscript. Please attend to each item and ensure your manuscript is fully compliant. If your revised manuscript is not aligned with these requests on major issues, such as those concerning statistics, it may be returned to you for further revisions without re-review.

Please submit the following items:

- Revised manuscript
- Point-by-point response to the referees' comments

- Cover letter (as a separate document)
- <https://www.nature.com/documents/nr-reporting-summary.pdf>>Nature Research Reporting Summary
- Completed Editorial Request Table (attached).

via this link: Link Redacted .

Additional guidance is available in our style and formatting guide Communications Psychology formatting guide.

Best regards,

Troy Lui

Troy Lui, PhD
Associate Editor
Communications Psychology

REVIEWER EXPERTISE:

Reviewer #1: memory/multisensory integration, EEG

Reviewer #2: memory/multisensory integration, entrainment, EEG

Reviewer #3: memory/multisensory integration, entrainment, EEG

REVIEWER REPORTS:

Reviewer #1 (Remarks to the Author):

The authors investigated the influence of pre-stimulus MEG alpha power on theta rhythmic sensory stimulation (RSS) and associated enhancement of episodic memory. The results revealed that pre-stimulus alpha predicted the strength of entrainment in sensory brain regions, and this entrainment was associated with episodic memory formation.

The manuscript is well-written, and the topic is interesting. However, I identified some issues with the study as presented which I have listed below:

1. The study failed to replicate the previously observed overall relationship between rhythmic sensory stimulation and episodic memory. This is in line with findings from other labs which have also failed to observe this effect. It is surprising, given that the study included a replication, that no power analysis is reported. Based on previous effect sizes, was the current study sufficiently powered to detect such effects?
2. If an overall RSS effect was obscured by trial-by-trial variability in the strength of entrainment, how could previous studies which did not account for this have detected the effect? It struck me on reading the manuscript that uncertainty over the veracity of the originally observed relationship between theta rhythmic sensory stimulation and episodic memory performance would best be clarified by a highly powered, pre-registered direct replication. The sample size in the current study (25 after exclusion) seems low and the analyses beyond the overall effect (i.e., those grouping trials by observed versus expected phase differences) are highly exploratory. In other words, it is disconcerting that the overall original effect didn't replicate, and I'm not convinced that the results of the subsequently presented analyses satisfactorily explain the failed replication.

3. For the pre-stimulus alpha analyses, it is mentioned in the methods that baseline correction was performed. Can the authors clarify exactly what this involved as I was a bit confused by the description (it seemed like post-stimulus activity was being used as a baseline for pre-stimulus?). For pre-stimulus activity, why is any baseline correction needed at all? Why was between 0.7 and -0.2 s before stimulus onset chosen as the time-period of interest?

4. Also, regarding the pre-stim alpha analysis, was a one-tailed test used to compare alpha between the mismatch and match conditions? If so, why?

Reviewer #2 (Remarks to the Author):

This manuscript investigates how variability in audiovisual theta entrainment predicts associative memory and identifies pre-stimulus alpha as a candidate brain-state mechanism underlying this variability. While the primary goal is to test the TIME framework, the study advances it by showing that memory facilitation is state-dependent and critically shaped by ongoing attentional fluctuations indexed by alpha. Overall, I found this to be a strong and well-designed study that makes a valuable contribution to the literature on multisensory entrainment and memory. My comments below are intended to help clarify interpretation and ensure the analyses align as closely as possible with the theoretical claims.

Major points

1. The manuscript already makes clear that there is no significant mean-level memory difference between 0° and 180°, and that the TIME effect only emerges when trial-level entrainment is modeled (phase × memory interaction). This state-dependent framing is important and I think it would help to emphasize it even more explicitly in the Discussion. In particular, the null replication reported by Serin and colleagues (2024) provides useful context: they observed no group-level benefit of synchronous theta, and even worse memory than in a no-flicker control. One procedural difference is that their study presented conditions in separate blocks, whereas the present study intermixed 0° and 180° trials. Blocking may have stabilized participants' expectations and reduced variability, while the intermixed design here allowed trial-by-trial fluctuations in entrainment and pre-stimulus alpha to emerge. Highlighting this contrast could help readers understand why Serin's results diverged and how the present trial-level approach provides a mechanistic account of such negative findings.

2. Another point concerns the interpretation of pre-stimulus alpha. The effect is currently described in general attentional terms, but it is not fully clear whether you see this more as a reflection of vigilance/arousal, as selective attentional gating, or perhaps as an interaction where arousal boosts the efficiency of attentional gating. In principle, pupil diameter and fixation stability from your Eyelink recordings could provide converging evidence and the best way to arbitrate between these interpretations would be a new analysis of those signals. That said, I do not see this as required for the present paper: even a short theoretical discussion of how you interpret the alpha effect in relation to vigilance and attention would already help clarify the psychological meaning of your results.

3. Reading about the 180° condition made me think of a question regarding its interpretation. It is framed as "out-of-phase at 4 Hz," but one could also view it differently: each modality is still driven at 4 Hz, yet their peaks alternate. From a supramodal perspective (e.g., attentional control networks), this alternation might in principle generate an 8 Hz rhythm of alternating salience across modalities. The classical TIME framework is already supramodal in nature, since memory benefits arise when auditory and visual streams are temporally aligned and bound together via higher-order networks. Prior work has shown that rhythmic stimulation in one modality can entrain or reset oscillations in networks supporting another modality (e.g., Lakatos et al., 2009) and that unimodal visual rhythms can entrain supramodal frontoparietal theta networks and enhance auditory working memory (e.g., Albouy et al., 2022). However, to my knowledge no paradigm has tested whether alternating audiovisual streams can induce a novel supramodal rhythm (e.g., 8 Hz). Would you consider exploring this possibility in your dataset, for example, by testing whether 8 Hz tagging emerges in PPC during 180° trials? Even exploratory analyses could be informative and I am genuinely curious whether this might help clarify why memory is impaired in 180°.

4. Another methodological point concerns frequency specificity. If I'm correct, for the main entrainment analyses the MEG signals were filtered broadly between 1.5 and 9 Hz before computing instantaneous phase differences. This makes it difficult to know whether the reported effects are truly specific to 4 Hz entrainment or partly reflect contributions from neighboring frequencies. I also noticed that you applied a narrower 3.5-4.5 Hz filter in the unimodal evoked power analyses but this was not extended to the cross-modal single-trial entrainment index. Would you consider re-running the key entrainment analyses with a narrow band centered on the stimulation frequency (e.g., 3.5-4.5 Hz)? This would provide stronger evidence that the observed memory effects are indeed driven by 4 Hz entrainment rather than by broadband low-frequency activity (and would also rule out the possibility of minor contributions from other frequencies, such as around 8 Hz if some supramodal resonance hypothetically exists).

5. Finally, regarding alpha baseline choice and normalization: pre-stimulus alpha was quantified in the -0.7 to -0.2 s window but normalized using a post-stimulus baseline (3.2-3.7 s). If I'm correct, this baseline falls immediately after the 3 s rhythmic stimulation within the same trial epoch. That period may still contain residual stimulation-related activity, which could bias the normalization of pre-stimulus alpha. Would you consider using a different, clearly stimulation-free pre-trial interval as baseline if available? This would avoid potential contamination and provide a stronger basis for interpreting low pre-stimulus

alpha as an endogenous attentional state.

Minor points

Several simple effects are reported one-sided. I might have missed the info but could you justify this approach (why not two-sided)?

Cluster-based tests are corrected, but other families of analyses such as ANOVAs, t-tests, and binning do not appear to be. Could you clarify how you approached correction for multiple comparisons in those cases?

The closing line of the Discussion suggests clinical applicability, which feels premature in my opinion as the paper do not investigate the long term effect of this kind of sensory stimulation.

It may also be useful to flag more clearly which analyses replicate earlier work (for instance, the back-sorting or “hockey-stick” phase binning) versus those that are exploratory (for example, PPC-hippocampus connectivity).

__ Thank you for your work.

Reviewer #3 (Remarks to the Author):

This manuscript examines how ongoing brain states, specifically prestimulus alpha oscillations, influence the effectiveness of theta rhythmic sensory stimulation (RSS) in supporting memory. The study builds on prior work from this group showing that phase alignment of theta oscillations across sensory regions facilitates binding in the hippocampus, and that RSS can enhance episodic memory when visual and auditory cortices are entrained in-phase (0°) compared to out-of-phase (180°).

Here, the authors replicate their earlier finding that RSS successfully entrains 4 Hz activity in visual and auditory cortices. While they do not observe memory differences between the 0° and 180° conditions, they show that entrainment strength predicts subsequent memory on a trial-by-trial basis. Importantly, prestimulus alpha power predicted theta entrainment strength, with lower alpha preceding trials that matched the expected phase offset, particularly in the 0° condition. Source analyses identified the posterior cingulate cortex (PCC) as a generator of prestimulus alpha, and prestimulus coupling between PCC and hippocampus further supported this link. Together, the results highlight that background brain states can shape RSS effects and should be considered in future work.

The study has several notable strengths. The theoretical basis is strong, and there is a clear rationale for testing how prestimulus oscillations contribute to intra-individual variability in entrainment and memory encoding. The methods are easy to follow and well established, having been used in prior work by this group and others. The authors also include several control analyses that strengthen their conclusions. The results are compelling and will be of broad interest for those studying sensory processing, multisensory integration, and memory encoding more generally, and those aiming to improve memory using RSS.

There are, however, a few areas where the manuscript could be strengthened. A deeper integration of the literature on alpha oscillations in attention would sharpen the theoretical framing. Expanding the discussion of PCC's role in attentional control and memory, as well as its interactions with the hippocampus, would also help contextualize the observed prestimulus PCC-hippocampal coupling results.

The authors note increased alpha power in the second half of the experiment but no corresponding differences in memory. It would be informative to test whether RSS strength differed across the task. Relatedly, I am curious about the possible connection between prestimulus alpha and habituation to the theta stimulus (resulting in weaker in entrainment). Are prestimulus alpha/attention related to sensory habituation?

Overall, I believe the results of this study will make an important contribution. However, minor revisions to expand the discussion of alpha oscillations, PCC, and habituation, would strengthen the theoretical impact.

Version 1:

Decision Letter:

Dear Dr Wang,

Your manuscript titled "Pre-stimulus alpha power modulates trial-by-trial variability in theta rhythmic multisensory entrainment strength and theta-induced memory effect" has now been seen by our reviewers, whose comments appear below. In light of their advice I am delighted to say that we are happy, in principle, to publish a suitably revised version in Communications Psychology.

We therefore invite you to revise your paper one last time to address the remaining concerns of our reviewers and a list of editorial requests. At the same time we ask that you edit your manuscript to comply with our format requirements and to maximise the accessibility and therefore the impact of your work.

EDITORIAL REQUESTS:

SUBMISSION INFORMATION:

OPEN ACCESS:

*** TRANSPARENT PEER REVIEW:** Communications Psychology uses a transparent peer review system. On author request, confidential information and data can be removed from the published reviewer reports and rebuttal letters prior to publication. If you are concerned about the release of confidential data, please let us know specifically what information you would like to have removed. Please note that we cannot incorporate redactions for any other reasons.

*** CODE AVAILABILITY:** All Communications Psychology manuscripts must include a section titled "Code Availability" at the end of the methods section. We require that the custom analysis code supporting your conclusions is made available in a publicly accessible repository at this stage; please choose a repository that generates a digital object identifier (DOI) for the code; the link to the repository and the DOI must be included in the Code Availability statement. Publication as Supplementary Information will not suffice.

*** DATA AVAILABILITY:**

Link Redacted

Best regards,

Troy Lui

Troy Lui, PhD
Associate Editor
Communications Psychology

REVIEWERS' COMMENTS:

Reviewer #1 (Remarks to the Author):

I thank the authors for taking my initial concerns seriously and altering the manuscript accordingly. I believe the manuscript has been strengthened. I note that the suggestion that the hardware used for delivering auditory stimulation in the scanner may have contributed to failure to replicate previous effects is highly speculative and would need to be empirically tested. However, I have no further comments on the manuscript.

Reviewer #2 (Remarks to the Author):

I am satisfied with the authors' responses and with the revisions made to the manuscript. The concerns raised in the previous round have been adequately addressed, and the manuscript is now suitable for publication in its current form. I thank the authors for their careful work and their important contribution to the field.

Reviewer #3 (Remarks to the Author):

I appreciate the authors' efforts to address my comments and those of the other reviewers. The authors have adequately addressed my prior comments. I have no additional feedback.

We would like to thank the reviewers for their insightful comments and the editor for overseeing the review process. Additional analyses have been conducted according to the reviewers' suggestions, which we find strengthened our findings. Please find below our detailed responses. We hope to have adequately addressed all reviewers' concerns. This response letter is formatted such that the reviewers' original comments appear in normal text, our responses are underlined, and the changed sections in the manuscript are in *italics*.

Reviewer #1 (Remarks to the Author):

The authors investigated the influence of pre-stimulus MEG alpha power on theta rhythmic sensory stimulation (RSS) and associated enhancement of episodic memory. The results revealed that pre-stimulus alpha predicted the strength of entrainment in sensory brain regions, and this entrainment was associated with episodic memory formation.

The manuscript is well-written, and the topic is interesting. However, I identified some issues with the study as presented which I have listed below:

1. The study failed to replicate the previously observed overall relationship between rhythmic sensory stimulation and episodic memory. This is in line with findings from other labs which have also failed to observe this effect. It is surprising, given that the study included a replication, that no power analysis is reported. Based on previous effect sizes, was the current study sufficiently powered to detect such effects?

Response: We thank the reviewer for the questions regarding the effect sizes, required sample size and power. We have conducted a power analysis using G*Power (Faul et al., 2009) based on the effect sizes averaged across all our previous studies (four studies from our lab), Cohen's $d = 0.765$. Total sample size 25 is required given $\alpha = 0.05$ (one-tailed paired sample t-test) and 90% statistical power. A sensitivity curve plotted on statistical power as a function of effect size (from 0.4 to 0.85 in a range of effect sizes in our previous studies) shown below suggests that we should still have 60% power for the smallest effect size among all our previous studies. Please note that when we conducted the power analysis based on our previous studies, Serin et al. (2024) had not been published yet. Therefore, the power analysis was only based on all the studies from our lab, which suggested that the current study was adequately powered. However, if the studies from Serin et al. (2024) were included in the power analysis, the mean effect sizes across seven studies (four from our lab and three from Serin et al. (2024)), Cohen's $d = 0.404$ would require total sample size 54 given $\alpha = 0.05$ (one-tailed paired sample t-test) and 90% statistical power. Then the current study is actually not adequately powered.

A statement on the required sample size has been added to the ‘Participants’ in the Materials and methods section in Page 18:

“A power analysis using G*Power⁵⁸ based on the effect sizes averaged across four previous studies from our lab, Cohen’s $d = 0.765$. Total sample size 25 is required given $\alpha = 0.05$ (one-tailed sample t-test) and 90% statistical power.”

2. If an overall RSS effect was obscured by trial-by-trial variability in the strength of entrainment, how could previous studies which did not account for this have detected the effect? It struck me on reading the manuscript that uncertainty over the veracity of the originally observed relationship between theta rhythmic sensory stimulation and episodic memory performance would best be clarified by a highly powered, pre-registered direct replication. The sample size in the current study (25 after exclusion) seems low and the analyses beyond the overall effect (i.e., those grouping trials by observed versus expected phase differences) are highly exploratory. In other words, it is disconcerting that the overall original effect didn’t replicate, and I’m not convinced that the results of the subsequently presented analyses satisfactorily explain the failed replication.

Response:

We appreciate the reviewer’s concern regarding the replication of the TIME effect. Based on effect sizes from our previous studies, our power analysis indicated that a sample size of 25 should be sufficient to detect a significant TIME effect. However, the current study did not replicate the effect at the average level. We believe this discrepancy may be attributable to the auditory stimulus delivery method required for MEG compatibility. As noted in the Discussion, the use of long, thin silicone tubes likely reduced the sound frequency bandwidth, introducing sensory noise that could increase variability in entrainment strength (Panzeri et al., 2010). This noise may also have impacted attentional engagement, which is known to modulate entrainment (Fisher et al., 2023).

Given these considerations, we investigated how variability in entrainment strength and attentional state might contribute to the observed variability in TIME. Importantly,

our approach to back-sorting trials based on actual phase differences is not novel or exploratory in isolation—it has been employed in prior EEG studies (Wang et al., 2018, 2023), where it significantly reduced behavioural variance. In our back-sorting analysis, we used the same parameters (e.g. filtering, source localised time-series analysis, etc.) that we have used in a previous EEG study (see Wang et al., 2018). Our findings therefore replicate the findings from previous studies, suggesting that accounting for trial-by-trial variability is a necessary refinement rather than an exploratory deviation.

We also note that a recent independent MEG study (Serin et al., 2024), which also delivered the audio stimuli through thin silicon tubes, similarly failed to replicate the TIME effect at the average level, despite methodological differences. This further supports the notion that MEG-specific auditory delivery systems may introduce variability that obscures the overall effect.

Finally, we agree that a highly powered, pre-registered direct replication would be valuable for clarifying the robustness of the TIME effect. However, our current findings contribute important insights into the conditions under which the effect may or may not emerge, and highlight the need to consider physiological variability in entrainment when interpreting null results. We highlighted the parts where we refer to our previous studies using the same back-sorting approach on Pages 5, 7, 8, 15 in the manuscript and added a paragraph of Limitations and the end of Discussion:

“Limitations

The sample size in the current study was determined based on the effect sizes averaged across previous studies from our lab. We note that if the studies from Serin et al.¹⁸ were included, it would require total sample size of 54 given $\alpha = 0.05$ (one-tailed paired sample t-test) and 90% statistical power. Although Serin et al.¹⁸ was not published while we conducted the current study, a highly powered, pre-registered direction replication would be valuable for clarifying the robustness of the TIME. However, the current study contributes important insights into the conditions which the effect may or may not emerge, and highlight the need to consider physiological variability in entrainment when interpreting null results.”

3. For the pre-stimulus alpha analyses, it is mentioned in the methods that baseline correction was performed. Can the authors clarify exactly what this involved as I was a bit confused by the description (it seemed like post-stimulus activity was being used as a baseline for pre-stimulus?). For pre-stimulus activity, why is any baseline correction needed at all? Why was between 0.7 and -0.2 s before stimulus onset chosen as the time-period of interest?

Response: The pre-stimulus alpha activity was normalised by the average baseline alpha activity of the condition itself because alpha activity can be affected by non-cognitive factors such as fatigue or trial order (Benwell et al., 2019). The time of interest (TOI) was picked between -0.7 and -0.2 s because it avoided the influences from the stimulus or the motor response. The pre-stimulus activity was normalised by the mean activity between 3.2 s and 3.7 s after stimulus onset across trials. Because the stimulus was 3 s long and the mean RT was 1.46 s. Therefore, on average the activity in this period did not involve stimulus offset response or motor evoked

response. This has been highlighted in the 'Pre-stimulus oscillatory activity analysis' in Materials and methods on Pages 25.

4. Also, regarding the pre-stim alpha analysis, was a one-tailed test used to compare alpha between the mismatch and match conditions? If so, why?

Response: We used indeed a one-tailed test to compare alpha power between match and mismatch trials. This was done following the well-established relationship between alpha power and attention, as well as memory performance. In essence, a plethora of studies demonstrated that alpha power decreases over attended hemifields and that it decreases for later remembered stimuli (see Hanslmayr et al., 2012; Jensen & Mazaheri, 2010; Klimesch et al., 2007). We have now added the rationale of testing only one direction to the section 'Pre-stimulus oscillatory activity analysis' in Materials and methods on Page 25:

"It is well established that decreases in pre-stimulus alpha power are linked to attention and predict better subsequent task performance (see⁶⁷⁻⁶⁹). Therefore, a one-sample permutation t-test with spatio-temporal clustering was conducted with 1000 permutations and a right-tail at alpha level 0.05, to compare if the condition difference was significantly higher than 0 for gradiometer sensors and magnetometer sensors, respectively."

Reviewer #2 (Remarks to the Author):

This manuscript investigates how variability in audiovisual theta entrainment predicts associative memory and identifies pre-stimulus alpha as a candidate brain-state mechanism underlying this variability. While the primary goal is to test the TIME framework, the study advances it by showing that memory facilitation is state-dependent and critically shaped by ongoing attentional fluctuations indexed by alpha. Overall, I found this to be a strong and well-designed study that makes a valuable contribution to the literature on multisensory entrainment and memory. My comments below are intended to help clarify interpretation and ensure the analyses align as closely as possible with the theoretical claims.

Major points

1. The manuscript already makes clear that there is no significant mean-level memory difference between 0° and 180°, and that the TIME effect only emerges when trial-level entrainment is modeled (phase × memory interaction). This state-dependent framing is important and I think it would help to emphasize it even more explicitly in the Discussion. In particular, the null replication reported by Serin and colleagues (2024) provides useful context: they observed no group-level benefit of synchronous theta, and even worse memory than in a no-flicker control. One procedural difference is that their study presented conditions in separate blocks, whereas the present study intermixed 0° and 180° trials. Blocking may have stabilized participants' expectations and reduced variability, while the intermixed design here allowed trial-by-trial fluctuations in entrainment and pre-stimulus alpha to emerge. Highlighting this contrast could help readers understand why Serin's results

diverged and how the present trial-level approach provides a mechanistic account of such negative findings.

Response: We thank the review for their insights on this important procedural change of the study by Serin et al. (2024). We have added a few sentences in the second paragraph of Discussion to highlight the difference and discussed how this could help better understand the lack of TIME on average in both studies on Page 15:

“One of procedural changes Serin et al.¹⁸ made in their study was to block the synchronous or asynchronous trials, instead of intermixing them. Although they have not conducted the single-trial analysis, it would be interesting to see if blocking the trials can also introduce variability in entrainment strength, thus enabling to understand to what extent this could explain the lack of TIME on average level.”

2. Another point concerns the interpretation of pre-stimulus alpha. The effect is currently described in general attentional terms, but it is not fully clear whether you see this more as a reflection of vigilance/arousal, as selective attentional gating, or perhaps as an interaction where arousal boosts the efficiency of attentional gating. In principle, pupil diameter and fixation stability from your Eyelink recordings could provide converging evidence and the best way to arbitrate between these interpretations would be a new analysis of those signals. That said, I do not see this as required for the present paper: even a short theoretical discussion of how you interpret the alpha effect in relation to vigilance and attention would already help clarify the psychological meaning of your results.

Response: We thank the reviewer for raising this point, because it triggered us to investigate more closely the potential relationship between eye-movements and the pre-stimulus alpha effects, which is currently a hot topic in the field. We fully took the reviewer’s comment on board and analysed the time-resolved fixation data from the Eyelink recordings between match and mismatch conditions in the same time window as in the pre-stimulus alpha analysis. Interestingly, we did not see a statistical difference in fixation in the same time window. We have changed the manuscript in various places to incorporate these results by adding a supplementary figure, and changing the Results, Discussion and Materials and methods sections as follows:

Results on Page 12:

“Moreover, recent studies have linked fluctuations in alpha power to fixation-related eye movements^{35,36}. Therefore, we have compared the fixation density between the mismatch and match conditions in the same pre-stimulus time window and showed that there was no significant difference (Figure S4).”

and Discussion on Page 17:

“Alternatively, recent studies have shown that alpha power increases when fixation is maintained whereas initiating eye movements to explore stimuli leads to a decrease in alpha power^{35,36}. The difference in alpha power between two experimental conditions could be the difference in visual exploration due to the complexity of the

stimuli. We have shown that this could not explain the pre-stimulus alpha power effect between mismatched and matched trials in the current study. The fixation density did not differ between conditions in the same pre-stimulus time window where the alpha power difference was found.”

The procedure of the analysis has been added to the Materials and methods on Page 25:

“The analysis of eye tracking data followed the procedures in^{35,36}. The raw eye tracking data recorded simultaneously with the MEG recordings was converted from voltage to pixel coordinates following the tutorial from the Fieldtrip (https://www.fieldtriptoolbox.org/getting_started/eyetracker/eyelink/#what-are-the-units-of-the-eye-tracker-data). Blinks were replaced by NaN if the values were smaller than -0.1 z-score of the time series data. The data was then epoched the same way as done to the MEG data. The gaze positions along the horizontal (x) and vertical (y) axes were binned into a 1000 x 1000-pixel grid. A 2D histogram was computed using the numpy module histogram2d. The histogram was then smoothed using a Gaussian filter from scipy’s ndimage module with a smoothing factor of 5. A peak bin was found after averaging the 2D gaze heatmaps across 23 participants excluding one participant whose eye-tracking data was not properly recorded. Time-resolved gaze density was computed for each trial and for each participant by computing the 2D histogram every 50 ms. The data along the vertical direction was averaged across 100 bins centred at the peak y bin, resulting in a data structure of time x horizontal position of the gaze density. Same one-sample permutation t-test was conducted with 1000 permutations with two-tail at alpha level 0.05 to compare the difference between the mismatch and match conditions. The time window was cropped at -0.7 and -0.2 s to be consistent with the pre-stimulus alpha time window. The horizontal pixels were restricted to 100 bins centred at the peak x coordinate.”

A supplementary figure has been added on Page 39:

A**B**
Figure S4. Gaze density in the match and mismatch conditions. *A*, gaze density map was computed for each participant across trials and average across participants. Red colour

suggests longer time spent on a location. The screen resolution was 1920 x 1080. The bin where maximum time spent on was around the centre of the screen. B, time-resolved histogram of gaze density around fixation in match (top) and mismatch (middle) conditions, and the difference between the mismatch and match conditions (bottom). The red colour indicates increased gaze density at the fixation before stimulus onset. This did not statistically differ between the two conditions.

3. Reading about the 180° condition made me think of a question regarding its interpretation. It is framed as “out-of-phase at 4 Hz,” but one could also view it differently: each modality is still driven at 4 Hz, yet their peaks alternate. From a supramodal perspective (e.g., attentional control networks), this alternation might in principle generate an 8 Hz rhythm of alternating salience across modalities. The classical TIME framework is already supramodal in nature, since memory benefits arise when auditory and visual streams are temporally aligned and bound together via higher-order networks. Prior work has shown that rhythmic stimulation in one modality can entrain or reset oscillations in networks supporting another modality (e.g., Lakatos et al., 2009) and that unimodal visual rhythms can entrain supramodal frontoparietal theta networks and enhance auditory working memory (e.g., Albouy et al., 2022). However, to my knowledge no paradigm has tested whether alternating audiovisual streams can induce a novel supramodal rhythm (e.g., 8 Hz). Would you consider exploring this possibility in your dataset, for example, by testing whether 8 Hz tagging emerges in PPC during 180° trials? Even exploratory analyses could be informative and I am genuinely curious whether this might help clarify why memory is impaired in 180°.

Response: We thank the reviewer’s suggestion to check if an 8 Hz signal emerges in the PPC in the 180° phase offset condition. We have conducted the same LCMV source reconstruction analysis to extract time series from the PPC peak voxel where the strongest pre-stimulus alpha power effect was found between the mismatch and match conditions. We then performed a time-frequency analysis on the PPC time series for each phase offset condition and conducted a cluster-base permutation test to test the statistical difference in frequency of interest 7.5 and 8.5 Hz and time of interest 0.75 and 2.75 s. We did not find any statistical evidence that the 8 Hz power in the 180° phase offset condition was significantly stronger than in the 0° phase offset condition. Therefore, we decided not to include this result in the manuscript to keep the manuscript streamlined and not clutter it with too much information that is not central to our story. Please see the figure below:

4. Another methodological point concerns frequency specificity. If I'm correct, for the main entrainment analyses the MEG signals were filtered broadly between 1.5 and 9 Hz before computing instantaneous phase differences. This makes it difficult to know whether the reported effects are truly specific to 4 Hz entrainment or partly reflect contributions from neighboring frequencies. I also noticed that you applied a narrower 3.5-4.5 Hz filter in the unimodal evoked power analyses but this was not extended to the cross-modal single-trial entrainment index. Would you consider re-running the key entrainment analyses with a narrow band centered on the stimulation frequency (e.g., 3.5-4.5 Hz)? This would provide stronger evidence that the observed memory effects are indeed driven by 4 Hz entrainment rather than by broadband low-frequency activity (and would also rule out the possibility of minor contributions from other frequencies, such as around 8 Hz if some supramodal resonance hypothetically exists).

Response: We bandpass filtered the single trial time series between 1.5 and 9 Hz before Hilbert transformation. The justification of selecting this frequency band is to be consistent with our previous study that did the same single trial analysis (Wang et al. 2018). We did not use a narrow frequency band because filtering in a very narrow band might overestimate the stationarity of the signal, i.e. artificially return highly

stationary sine wave signals. Capturing the dynamic and variability in phase modulation was especially important to the single trial phase bin sorting analysis. Therefore, the set of filters we chose could provide a balance between capturing the modulation frequency and the dynamic in phase modulation, and on top of that are consistent with our previous study, which is important in terms of replicability (see Reviewer 1's comment #2).

5. Finally, regarding alpha baseline choice and normalization: pre-stimulus alpha was quantified in the -0.7 to -0.2 s window but normalized using a post-stimulus baseline (3.2-3.7 s). If I'm correct, this baseline falls immediately after the 3 s rhythmic stimulation within the same trial epoch. That period may still contain residual stimulation-related activity, which could bias the normalization of pre-stimulus alpha. Would you consider using a different, clearly stimulation-free pre-trial interval as baseline if available? This would avoid potential contamination and provide a stronger basis for interpreting low pre-stimulus alpha as an endogenous attentional state.

Response: We thank the reviewer for noticing the potential issue of the baseline window, which was also raised by reviewer 1. We selected the baseline window starting from 0.2 s after stimulus offset to avoid the offset potentials. On average the activity in this period should not involve stimulus offset response. We have clearly stated this on Page 25:

“Since the stimulus length was 3 s, on average the activity after 3.2 s should not involve stimulus offset response.”

Minor points

Several simple effects are reported one-sided. I might have missed the info but could you justify this approach (why not two-sided)?

Response: The simple effects where one-sided tests were used should all be the replication analyses. Therefore, we had clear hypotheses about those tests. This has been highlighted in ‘Statistical analysis of MEG source reconstructed data’ in the Materials and methods on Page 23:

“Since the analyses were replication of our previous study, all alternative hypotheses of the t-tests were one-sided.”

Cluster-based tests are corrected, but other families of analyses such as ANOVAs, t-tests, and binning do not appear to be. Could you clarify how you approached correction for multiple comparisons in those cases?

Response: In our analysis, we did not apply corrections to the subsidiary pairwise comparisons because these were only conducted following a significant interaction effect. This approach is grounded in the principle that post hoc comparisons are justified when guided by a significant omnibus test, and in such cases, each pairwise comparison is hypothesis-driven rather than exploratory. We have clarified this rationale in the revised manuscript ‘Statistical analysis of MEG source reconstructed

data' in the Materials and methods on Page 23, to ensure transparency in our analytical approach:

“The same pairwise comparisons were tested using paired sample t-tests as done in the previous study¹¹ if a significant interaction was shown.”

The closing line of the Discussion suggests clinical applicability, which feels premature in my opinion as the paper do not investigate the long term effect of this kind of sensory stimulation.

Response: We agree with the reviewer that the current study did not investigate the long-term effect of the rhythmic sensory stimulation (RSS), thus it would be premature to suggest any clinical applicability. However, the RSS has been used for clinical trials (Chan et al., 2022). We hope to provide better understanding on the nature of the (null) effect induced by the RSS so that the future studies could take advantage to adjust the stimulation parameters and improve the stimulation protocols before using it for clinical populations.

It may also be useful to flag more clearly which analyses replicate earlier work (for instance, the back-sorting or “hockey-stick” phase binning) versus those that are exploratory (for example, PPC-hippocampus connectivity).

Response: We thank the reviewer for raising this point and agree that this should be highlight. It is also an issue that came up in response to Reviewer 1s concerns. As a result, we now highlight in the Results and Discussion on which analyses were replication (Pages 5, 8, 15) and which were exploratory (Page 11).

Thank you for your work.

Reviewer #3 (Remarks to the Author):

This manuscript examines how ongoing brain states, specifically prestimulus alpha oscillations, influence the effectiveness of theta rhythmic sensory stimulation (RSS) in supporting memory. The study builds on prior work from this group showing that phase alignment of theta oscillations across sensory regions facilitates binding in the hippocampus, and that RSS can enhance episodic memory when visual and auditory cortices are entrained in-phase (0°) compared to out-of-phase (180°).

Here, the authors replicate their earlier finding that RSS successfully entrains 4 Hz activity in visual and auditory cortices. While they do not observe memory differences between the 0° and 180° conditions, they show that entrainment strength predicts subsequent memory on a trial-by-trial basis. Importantly, prestimulus alpha power predicted theta entrainment strength, with lower alpha preceding trials that matched the expected phase offset, particularly in the 0° condition. Source analyses identified the posterior cingulate cortex (PCC) as a generator of prestimulus alpha, and prestimulus coupling between PCC and hippocampus further supported this link. Together, the results highlight that background brain states can shape RSS effects and should be considered in future work.

The study has several notable strengths. The theoretical basis is strong, and there is

a clear rationale for testing how prestimulus oscillations contribute to intra-individual variability in entrainment and memory encoding. The methods are easy to follow and well established, having been used in prior work by this group and others. The authors also include several control analyses that strengthen their conclusions. The results are compelling and will be of broad interest for those studying sensory processing, multisensory integration, and memory encoding more generally, and those aiming to improve memory using RSS.

There are, however, a few areas where the manuscript could be strengthened. A deeper integration of the literature on alpha oscillations in attention would sharpen the theoretical framing. Expanding the discussion of PCC's role in attentional control and memory, as well as its interactions with the hippocampus, would also help contextualize the observed prestimulus PCC-hippocampal coupling results. The authors note increased alpha power in the second half of the experiment but no corresponding differences in memory. It would be informative to test whether RSS strength differed across the task. Relatedly, I am curious about the possible connection between prestimulus alpha and habituation to the theta stimulus (resulting in weaker in entrainment). Are prestimulus alpha/attention related to sensory habituation?

Overall, I believe the results of this study will make an important contribution. However, minor revisions to expand the discussion of alpha oscillations, PCC, and habituation, would strengthen the theoretical impact.

Response: We thank the reviewer for their suggestion to expand the discussion on the role of alpha oscillations, PPC and analysing the entrainment strength across the task. We followed both suggestions. Concerning the latter, we analysed whether entrainment strength changes over time as a function of subsequent memory and pre-stimulus alpha power. To this end, we computed entrainment strength for subsequently remembered and forgotten trials in each phase offset condition. Then we averaged the entrainment strength across those trials depending on their trial order and conducted a 2 (phase offset condition: 0° vs 180°) x 2 (trial order: 1st vs 2nd half in the trials) x 2 (subsequent memory: remembered vs forgotten) Repeated Measures ANOVA. A significant interaction between phase offset condition and subsequent memory was found, which is consistent with our analysis on the subsequent memory effect of entrainment strength. Although there was a trend that the 1st half trials' entrainment strength was stronger than the 2nd half trials', the main effect of trial order was not statistically significant, $F(1, 23) = 3.299, p = 0.082$. This has been added to the second last paragraph in the Discussion on Page 17:

“Further, although it was not statistically significant, a trend showed that the first half trials' entrainment strength was stronger than the second half trials’.”

We have added a paragraph to expand the discussion on the role of alpha oscillations and PPC at the end of the third paragraph of the Discussion on Page 16:

“A proposed mechanism of the PPC alpha oscillations is to gate the information flow, which increases the engagement of the contralateral dorsal attention network that is associated with a target and reduce the information flow in the ipsilateral side of the distractor⁴⁷. Such a gating mechanism could serve resource allocation in the cortical areas⁴⁸. This is also consistent with the findings in episodic memory formation and retrieval, which suggest that alpha power decreases reflect richer representation of information^{49,50}. Therefore, the stronger theta phase coupling between the PPC and the hippocampus shown in the current study could reflect that the sensory information is more likely to reach the hippocampus.”

References:

- Benwell, C. S. Y., London, R. E., Tagliabue, C. F., Veniero, D., Gross, J., Keitel, C., & Thut, G. (2019). Frequency and power of human alpha oscillations drift systematically with time-on-task. *NeuroImage*, *192*, 101–114.
<https://doi.org/10.1016/j.neuroimage.2019.02.067>
- Chan, D., Suk, H.-J., Jackson, B. L., Milman, N. P., Stark, D., Klerman, E. B., Kitchener, E., Avalos, V. S. F., Weck, G. de, Banerjee, A., Beach, S. D., Blanchard, J., Stearns, C., Boes, A. D., Uitermarkt, B., Gander, P., Iii, M. H., Sternberg, E. J., Nieto-Castanon, A., ... Tsai, L.-H. (2022). Gamma frequency sensory stimulation in mild probable Alzheimer’s dementia patients: Results of feasibility and pilot studies. *PLOS ONE*, *17*(12), e0278412.
<https://doi.org/10.1371/journal.pone.0278412>
- Fisher, V. L., Dean, C. L., Nave, C. S., Parkins, E. V., Kerkhoff, W. G., & Kwakye, L. D. (2023). Increases in sensory noise predict attentional disruptions to audiovisual speech perception. *Frontiers in Human Neuroscience*, *16*.
<https://doi.org/10.3389/fnhum.2022.1027335>
- Hanslmayr, S., Staudigl, T., & Fellner, M.-C. (2012). Oscillatory power decreases and long-term memory: The information via desynchronization hypothesis.

Frontiers in Human Neuroscience, 6.

<https://www.frontiersin.org/articles/10.3389/fnhum.2012.00074>

Jensen, O., & Mazaheri, A. (2010). Shaping Functional Architecture by Oscillatory Alpha Activity: Gating by Inhibition. *Frontiers in Human Neuroscience*, 4.

<https://www.frontiersin.org/articles/10.3389/fnhum.2010.00186>

Klimesch, W., Sauseng, P., & Hanslmayr, S. (2007). EEG alpha oscillations: The inhibition–timing hypothesis. *Brain Research Reviews*, 53(1), 63–88.

<https://doi.org/10.1016/j.brainresrev.2006.06.003>

Panzeri, S., Brunel, N., Logothetis, N. K., & Kayser, C. (2010). Sensory neural codes using multiplexed temporal scales. *Trends in Neurosciences*, 33(3), 111–120.

<https://doi.org/10.1016/j.tins.2009.12.001>

Serin, F., Wang, D., Davis, M. H., & Henson, R. (2024). Does theta synchronicity of sensory information enhance associative memory? Replicating the theta-induced memory effect. *Brain and Neuroscience Advances*, 8,

23982128241255798. <https://doi.org/10.1177/23982128241255798>

Wang, D., Clouter, A., Chen, Q., Shapiro, K. L., & Hanslmayr, S. (2018). Single-Trial Phase Entrainment of Theta Oscillations in Sensory Regions Predicts Human Associative Memory Performance. *The Journal of Neuroscience*, 38(28),

6299–6309. <https://doi.org/10.1523/JNEUROSCI.0349-18.2018>

Wang, D., Shapiro, K. L., & Hanslmayr, S. (2023). Altering stimulus timing via fast rhythmic sensory stimulation induces STDP-like recall performance in human episodic memory. *Current Biology*, 33(15), 3279–3288.e7.

<https://doi.org/10.1016/j.cub.2023.06.062>